# A neural circuit for gamma-band coherence across the retinotopic map in mouse visual cortex

Richard Hakim[1,2†], Kiarash Shamardani[1,2], Hillel Adesnik[1,2*]

[1]Department of Molecular and Cell Biology, University of California, Berkeley, United States; [2]Helen Wills Neuroscience Institute, University of California, Berkeley, United States

**Abstract** Cortical gamma oscillations have been implicated in a variety of cognitive, behavioral, and circuit-level phenomena. However, the circuit mechanisms of gamma-band generation and synchronization across cortical space remain uncertain. Using optogenetic patterned illumination in acute brain slices of mouse visual cortex, we define a circuit composed of layer 2/3 (L2/3) pyramidal cells and somatostatin (SOM) interneurons that phase-locks ensembles across the retinotopic map. The network oscillations generated here emerge from non-periodic stimuli, and are stimulus size-dependent, coherent across cortical space, narrow band (30 Hz), and depend on SOM neuron but not parvalbumin (PV) neuron activity; similar to visually induced gamma oscillations observed in vivo. Gamma oscillations generated in separate cortical locations exhibited high coherence as far apart as 850 μm, and lateral gamma entrainment depended on SOM neuron activity. These data identify a circuit that is sufficient to mediate long-range gamma-band coherence in the primary visual cortex.

DOI: https://doi.org/10.7554/eLife.28569.001

*For correspondence:
hadesnik@berkeley.edu

Present address: †Department of Neurobiology, Harvard Medical School, Massachusetts, United States

Competing interests: The authors declare that no competing interests exist.

## Introduction

Neural oscillations are ubiquitous features of brain activity and have been correlated with a diverse range of sensory, cognitive, and motor functions (*Buzsáki and Draguhn, 2004*; *Jia and Kohn, 2011*). It has been proposed that gamma oscillations (20–80 Hz) within the primary visual cortex (V1) synchronize the activity of functionally related neural ensembles, resulting in their selective amplification in downstream visual areas (*Fries, 2009*; *Fries et al., 2007*; *Singer and Gray, 1995*). In cats, gamma oscillations recorded in two spatially separated regions in V1 become phase-locked when a continuous visual object occupies their corresponding receptive fields (*Gray et al., 1989*; *Gray and Singer, 1989*). In primates, studies have observed strong correlations between gamma power and visual attention (*Fries et al., 2001*), as well as increased gamma-band synchrony among visual cortical areas during visual tasks, suggesting a role for gamma oscillations in visual perception and selective attention (*Siegel et al., 2008*). However, the precise function of gamma rhythms in visual processing remains controversial, with some studies providing evidence for a role in perceptual coding (*Fries, 2009*; *Pritchett et al., 2015*; *Salinas and Sejnowski, 2001*; *Singer and Gray, 1995*; *Varela et al., 2001*; *Womelsdorf et al., 2007*) and others arguing that neural codes based on the synchrony or phase of spike times during gamma oscillations are unlikely to contribute to visual processing (*Ray and Maunsell, 2010*; *Shadlen and Movshon, 1999*; *Thiele and Stoner, 2003*). Since the cellular and circuit mechanisms underlying long-distance gamma-band coherence are unknown, the causal manipulations that could assess whether gamma rhythms are required for key aspects of perception and cognition have yet to be defined.

Mechanistically, many different kinds of neural circuits may contribute to gamma-band synchronization (*Bartos et al., 2007*; *Buzsáki and Wang, 2012*; *Mann and Paulsen, 2005*; *Whittington and Traub, 2003*). In the mammalian forebrain, gamma oscillation generation is typically thought to depend on the reciprocal interaction between nearby recurrently connected excitatory and inhibitory neurons, with the precise timing of oscillations depending on the rapid recruitment of inhibitory neurons to transiently and periodically suppress local network activity (*Buzsáki and Wang, 2012*; *Cardin, 2016*; *Jefferys et al., 1996*; *Mann et al., 2005*; *Tiesinga and Sejnowski, 2009*). Several studies have correlated the activity of individual subtypes of inhibitory cortical neurons with gamma-paced firing, finding, for instance, that fast-spiking parvalbumin-positive (PV) basket cells phase-lock particularly strongly to gamma-band activity in the local field potential (*Bartos et al., 2007*; *Buzsáki, 2004*; *Csicsvari et al., 2003*; *Gulyás et al., 2010*; *Hasenstaub et al., 2005*; *Salkoff et al., 2015*; *Siegle et al., 2014*). Other studies have demonstrated that different or multiple inhibitory subtypes are involved in gamma generation and coherence (*Craig and McBain, 2015*; *Kipiani, 2009*; *Takada et al., 2014*; *Vierling-Claassen et al., 2010*; *Whittington et al., 2011*), and that gap junctions might also contribute (*Ainsworth et al., 2011*; *Long et al., 2005*; *Szabadics et al., 2001*; *Traub et al., 2001*). Recently, it was also demonstrated that, in mouse primary visual cortex, somatostatin-positive (SOM) interneurons contribute to stimulus-size dependent, visually induced oscillations and coherence (*Chen et al., 2017*; *Veit et al., 2017*). Moreover, theoretical work has also suggested that dendrite-targeting interneurons might contribute to gamma-band coherence in the hippocampus (*Tort et al., 2007*).

Despite this work, the cellular and synaptic basis for gamma coherence across the retinotopic map in the visual cortex is poorly understood. This may be due to technical challenges associated with manipulating and recording the activity of neuronal ensembles separated by large distances with the sufficient spatial and temporal control needed to probe the underlying mechanisms of neural synchronization. Previously, we and others have shown that optogenetic stimulation of L2/3 pyramidal cells (PCs) induces powerful gamma-band oscillations in cortical slices and in vivo (*Adesnik and Scanziani, 2010*; *Shao et al., 2013*; *Takada et al., 2014*). In this study, we employed patterned illumination and bidirectional optogenetics to precisely control the generation and entrainment of gamma rhythms across the retinotopic map of the mouse primary visual cortex.

## Results

### Long-range coherence of photo-induced gamma rhythms

We took advantage of the ability to reliably and potently generate gamma oscillations by optogenetically stimulating PCs in L2/3 of cortical slices (*Adesnik and Scanziani, 2010*, *2012*; *Shao et al., 2013*; *Takada et al., 2014*). ChR2 was transfected into a sparse subset of L2/3 PCs in V1 using *in utero* electroporation (*Saito and Nakatsuji, 2001*), and a slow ramp of blue light targeted to L2/3 was used to reliably drive oscillatory network activity. Under these conditions, ChR2 expression is restricted to excitatory neurons (*Figure 1—figure supplement 1*) (*Adesnik and Scanziani, 2010*), and therefore all optogenetically evoked inhibition is driven polysynaptically through the network, rather than being of monosynaptic origin. Consistent with prior work in both S1 and V1, wide-field illumination of L2/3 generates strong gamma rhythms in excitatory and inhibitory currents measured in L2/3 cortical neurons (*Figure 1A,B*). To gain control over the spatial profile of excitation, we built and characterized a digital-micromirror-device (DMD) based illumination system that generates arbitrary multicolor light patterns with high spatial and temporal precision (*Figure 1—figure supplement 2*, *Figure 4—figure supplement 1*). Using this system, we found that the power of the gamma oscillations depended on the area of illumination, reminiscent of the dependence of gamma oscillations on visual stimulus size in vivo (*Gieselmann and Thiele, 2008*; *Jia et al., 2013*; *Ray et al., 2013*; *Veit et al., 2017*) (*Figure 1C*. Analyzed from 0 to 1000 ms post-stimulus onset.).

To probe the mechanisms of long-range gamma coherence across the retinotopic map, we examined if photo-induced gamma rhythms in L2/3 are phase-locked across distant ensembles. Using the DMD, we generated two patches of blue light, separated by distances ranging from 275 to 850 μm, to activate two separate regions in L2/3 of V1. This was done while making simultaneous intracellular recordings from two ChR2-negative PCs located at the center of each patch of light (*Figure 1D–F*). Photo-stimulation generated gamma oscillations in each illuminated region, as measured by large

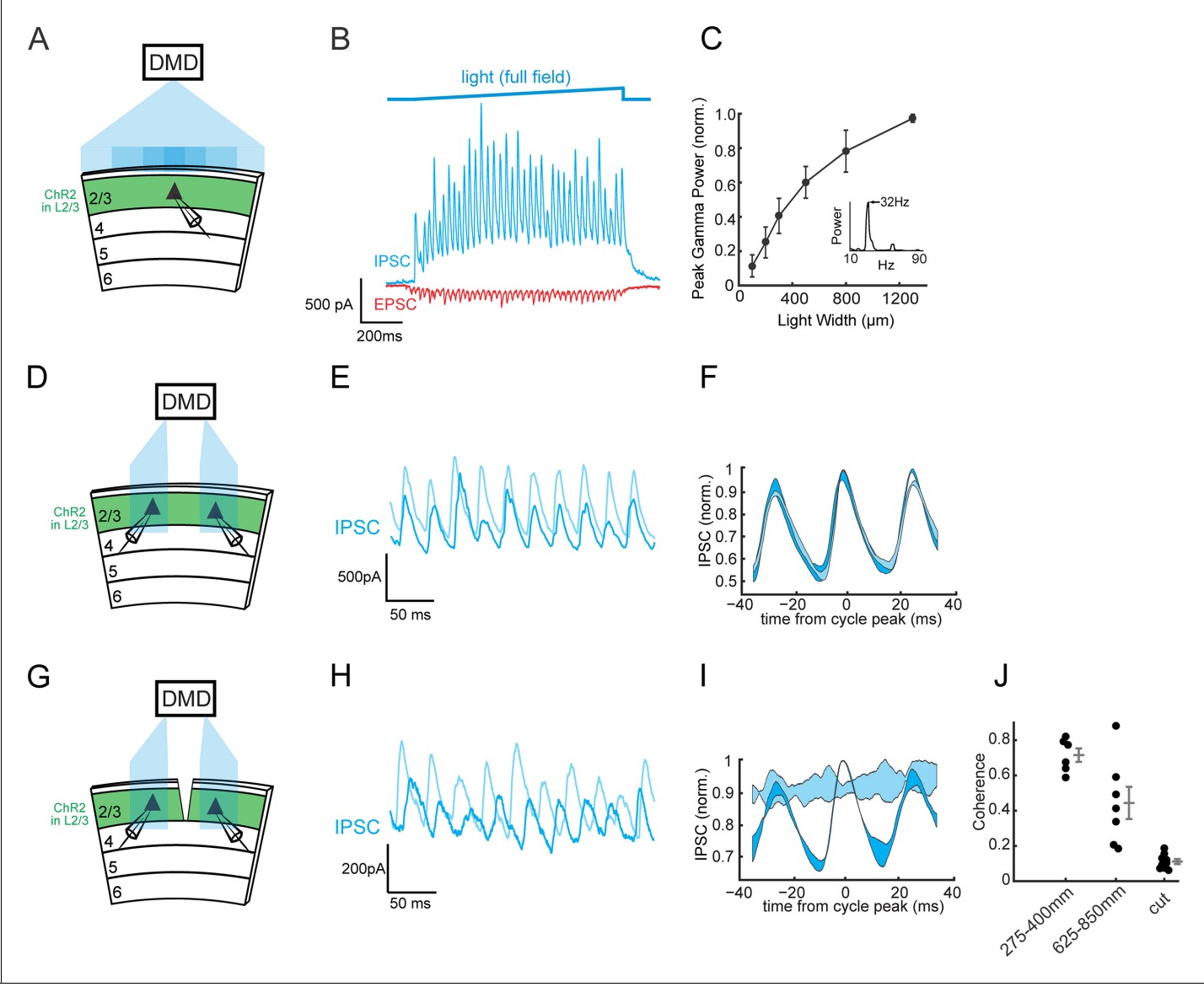

**Figure 1.** Horizontal circuits recruit local SOM interneurons to synchronize distant gamma generators. (**A**) Experimental schematic: A ChR2-negative Pyramidal cell is recorded in L2/3 of V1 while other ChR2-expressing L2/3 neurons are photo-stimulated with different sizes of blue light stimuli using a digital-micromirror-device (DMD). (**B**) Top: Time course of the light stimulus intensity (final intensity 1.1 mW/mm$^2$, see Materials and methods). Bottom: Example traces of voltage-clamped excitatory postsynaptic current (EPSC, red) and inhibitory postsynaptic current (IPSC, blue) during photo-induced gamma rhythms in V1. (**C**) Plot of peak gamma power versus the width of the photo-stimulus on L2/3 (n = 8, p<10$^{-4}$, Kruskal-Wallis ANOVA). Errorbars are s.e.m. (**D**) Experimental schematic: two ChR2-negative L2/3 pyramidal cells are simultaneously recorded while nearby ChR2-expressing L2/3 PCs are focally activated with separate blue light patches using a digital micro-mirror device (DMD). The distance between the blue light patches ranged from 275 to 850 μm (see **Figure 1—figure supplement 1B**). (**E**) Example traces of the voltage-clamped IPSCs from a pair of simultaneously recorded L2/3 PCs during photo-induction of two separate gamma oscillations. (**F**) Oscillation-triggered average of the IPSCs recorded in the pair in B) (triggered off the oscillations in one of the two cells, labeled in dark blue). Shading represents one standard deviation. (**G–I**) As in (**D–F**) but following a transection of L2/3 between the two recorded L2/3 PCs in transfected slices. (**J**) Scatter plot of the peak coherence of the oscillations in the two recorded neurons between the cut and the two intact conditions. Mean peak coherence with 275–400 μm separation (close): 0.72 ± 0.04, n = 6 pairs; mean peak coherence at 625–850 μm separation (far): 0.44 ± 0.09, n = 7 pairs; mean peak coherence at 275–400 μm with L2/3 cut (cut): 0.11 ± 0.01, n = 11 pairs; p<10$^{-3}$, Wilcoxon rank sum test between close and cut conditions; p<10$^{-3}$, Wilcoxon rank sum test between far and cut conditions. Errorbars are s.e.m.
DOI: https://doi.org/10.7554/eLife.28569.002

The following figure supplements are available for figure 1:

**Figure supplement 1.** *In utero* electroporation of ChR2-YFP into SOM-Cre, PV-Cre, and wild-type mice and spatial restriction of ChR2 expression to L2/3.

*Figure 1 continued on next page*

Figure 1 continued

DOI: https://doi.org/10.7554/eLife.28569.003

**Figure supplement 2.** Spatial resolution controls for DMD-based optogenetic activation of PCs.

DOI: https://doi.org/10.7554/eLife.28569.004

oscillatory currents recorded in each cell, and when both sites were co-stimulated, the two oscillations strongly phase-locked (*Figure 1D–F*). Coherence persisted even when the recorded cells were separated by up to 850 µm (*Figure 1J*), demonstrating that photo-induced gamma rhythms can synchronize over long distances.

L2/3 PCs project long-range horizontal axons that traverse the retinotopic map in V1 (*Gilbert and Wiesel, 1989*) and represent a plausible substrate for synchronizing gamma activity across spatially distant neural ensembles (*Gray et al., 1989*). To determine if horizontal axons in L2/3 are required for the long-range gamma phase-locking, we made a vertical cut between the two recorded cells, severing L2/3 horizontal axons. This transection spared gamma rhythm generation on either side of the cut, but abolished phase-locking between the two areas (*Figure 1G–J*; n = 11 pairs, p<10$^{-3}$, Wilcoxon rank sum test, see figure legend for statistics). These data demonstrate that lateral connections across L2/3 in V1 are necessary for long-range intralaminar gamma coherence.

## L2/3-induced gamma rhythm generation depends on somatostatin interneurons

To address the possible mechanisms of gamma rhythm induction, we probed the activity of the two major subclasses of cortical interneurons that provide inhibition to L2/3 PCs. Although substantial evidence implicates PV neurons in gamma entrainment (*Buzsáki and Wang, 2012*; *Cardin et al., 2009*; *Sohal et al., 2009*), we previously demonstrated that horizontal axons in L2/3 preferentially recruit SOM neurons in vitro (*Adesnik et al., 2012*), and that SOM neurons in vivo are also involved in visually induced gamma rhythms (*Veit et al., 2017*). Therefore, we targeted PV and SOM neurons to measure their activity during photo-induced gamma oscillations. To this end, we expressed ChR2 in PCs via *in utero* electroporation in SOM-Cre and PV-Cre mice crossed to a tdTomato Cre-reporter strain (*Figure 1—figure supplement 1*). This permitted photo-stimulation of L2/3 PCs, as in previous experiments, but further allowed simultaneous targeted electrophysiological recordings from tdTomato-expressing PV or SOM neurons. Strikingly, targeted loose-patch recordings from SOM and PV neurons in brain slices demonstrated that SOM neurons were reliably driven and tightly correlated with the photo-induced gamma rhythm in L2/3 (*Figure 2A–C*, n = 10 cells, mean induced rate: 25 ± 5 Hz. Data analyzed from 0 to 1000 ms post-stimulation onset.). Under the same conditions, the majority of L2/3 PV neurons fired at comparatively low rates or did not fire at all (4 ± 2 Hz, p=0.003, Wilicoxon rank sum test). To probe the basis for this unexpected result, whole-cell recordings from SOM neurons were made and demonstrated that they receive powerful, rhythmic excitation but very little inhibition (*Figure 2D,E*). In contrast, PV neurons and PCs received much more synaptic inhibition, consistent with their known inputs from SOM cells (*Figure 2D,E*; see figure legend for statistics) (*Hioki et al., 2013*; *Pfeffer et al., 2013*). The large difference in their excitation/inhibition ratio provides a synaptic basis for the differential recruitment of SOM and PV neurons during photo-induced gamma.

To causally dissect the specific contributions of SOM neurons to photo-induced gamma rhythms, we optogenetically suppressed SOM neurons and measured the resulting impact on gamma-band synaptic activity. To achieve simultaneous activation of PCs and suppression of SOM neurons, we injected a Cre-dependent adeno-associated virus (AAV) driving a red-light activated neural silencer, eNpHR3.0, into the visual cortex of *in utero* electroporated SOM-Cre mice. During photo-stimulation of PCs to induce gamma rhythms with blue light, red light was added to silence SOM neurons (blue light from 0 to 1000 ms post-stimulation onset, red light from 250 to 750 ms; analysis conducted from 250 to 750 ms). Local suppression of SOM neurons strongly reduced the inhibitory gamma power observed in PCs during photo-induced gamma activity (*Figure 3A,B* mean power reduction: 74 ± 3.5%, p<10$^{-5}$, Wilcoxon signed-rank test). Suppressing SOM cells also desynchronized excitatory synaptic input, demonstrating that their output is critical for pacing the activity of the excitatory network in the gamma band (*Figure 3B,C*; n = 8 cells, mean power reduction: 63 ± 7%,

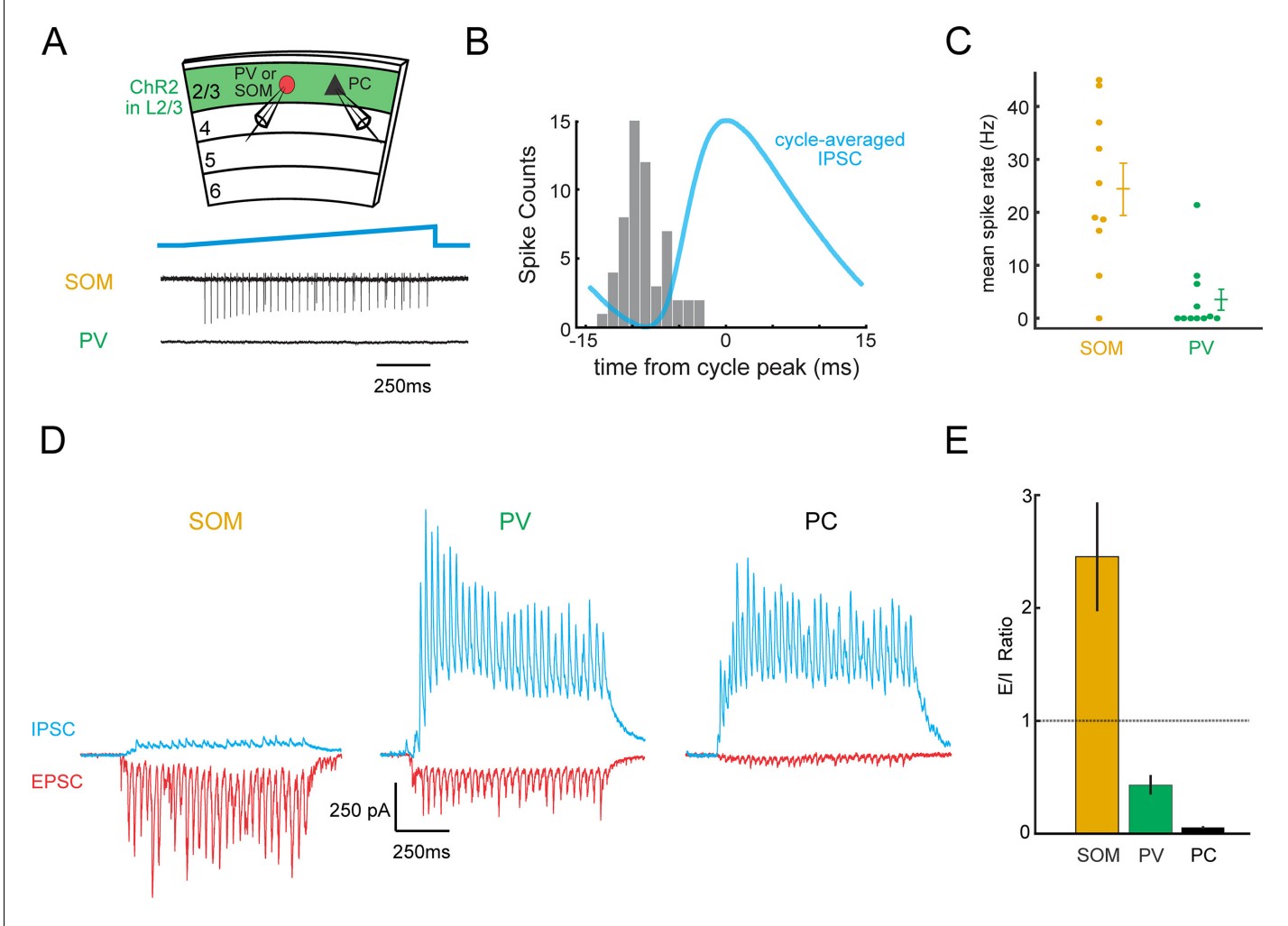

**Figure 2.** L2/3 gamma rhythms recruit SOM inhibitory neurons. (**A**) Top: Schematic of experiment. Loose-patch recordings are made from PV or SOM cells, and whole-cell recordings are made from Pyramidal cells (PC) to correlate to gamma rhythms. Slices expressing ChR2 in L2/3 and tdTomato in PV or SOM cells (see *Figure 1—figure supplement 1*). Bottom: example loose-patch recordings from a SOM and a PV neuron during photo-induced gamma. (**B**) Phase histogram of the spikes recorded in a SOM neuron during photo-induced gamma overlaid with the simultaneously recorded cycle-averaged IPSC. (**C**) Plot of the mean firing rate of PV and SOM neurons during photo-induced gamma (PV: n = 11 cells, SOM n = 10 cells, p<0.01, Wilcoxon rank sum test). Baseline spike rate was zero. (**D**) Representative traces of synaptic excitation (red) and inhibition (blue) from a SOM, PV, and PC. Baseline synaptic input is near-zero for both EPSCs and IPSCs. (**E**) Plot of the average E/I ratio in SOMs, PVs, and PCs during gamma activity in brain slices (n = 12 SOM cells, mean E/I ratio: 2.5 ± 0.5; 8 PV cells, mean E/I ratio: 0.43 ± 0.09; 10 pyramidal cells, mean E/I ratio: 0.054 ± 0.010; p<$10^{-3}$, Wilcoxon rank sum test between SOM and PV; p<$10^{-5}$, Kruskal-Wallis ANOVA). Error bars are s.e.m.

DOI: https://doi.org/10.7554/eLife.28569.005

p=0.004, Wilcoxon signed-rank test). In contrast, suppressing PV neurons had no significant effect on gamma power (*Figure 3—figure supplement 1A–E*, n = 9 cells, mean power reduction: 2.5 ± 6%, p=0.25, Wilcoxon signed-rank test), while it did significantly reduce the disynaptic IPSC to a brief pulse of blue light (*Figure 3—figure supplement 1F*, p=0.016, Wilcoxon signed-rank test). This demonstrates that gamma generation within L2/3 critically depends on the activity of SOM neurons.

These experiments still leave open the possibility that SOM neurons are only critical for maintaining gamma entrainment, but are not required at the very initiation of the photo-induced gamma oscillations, which might instead depend on PV neurons. This possibility is made more likely by studies showing that excitatory synapses onto PV neurons are strong but highly adapting, while those onto SOM cells are initially weak but highly facilitating (*Markram et al., 2004*; *Reyes et al., 1998*).

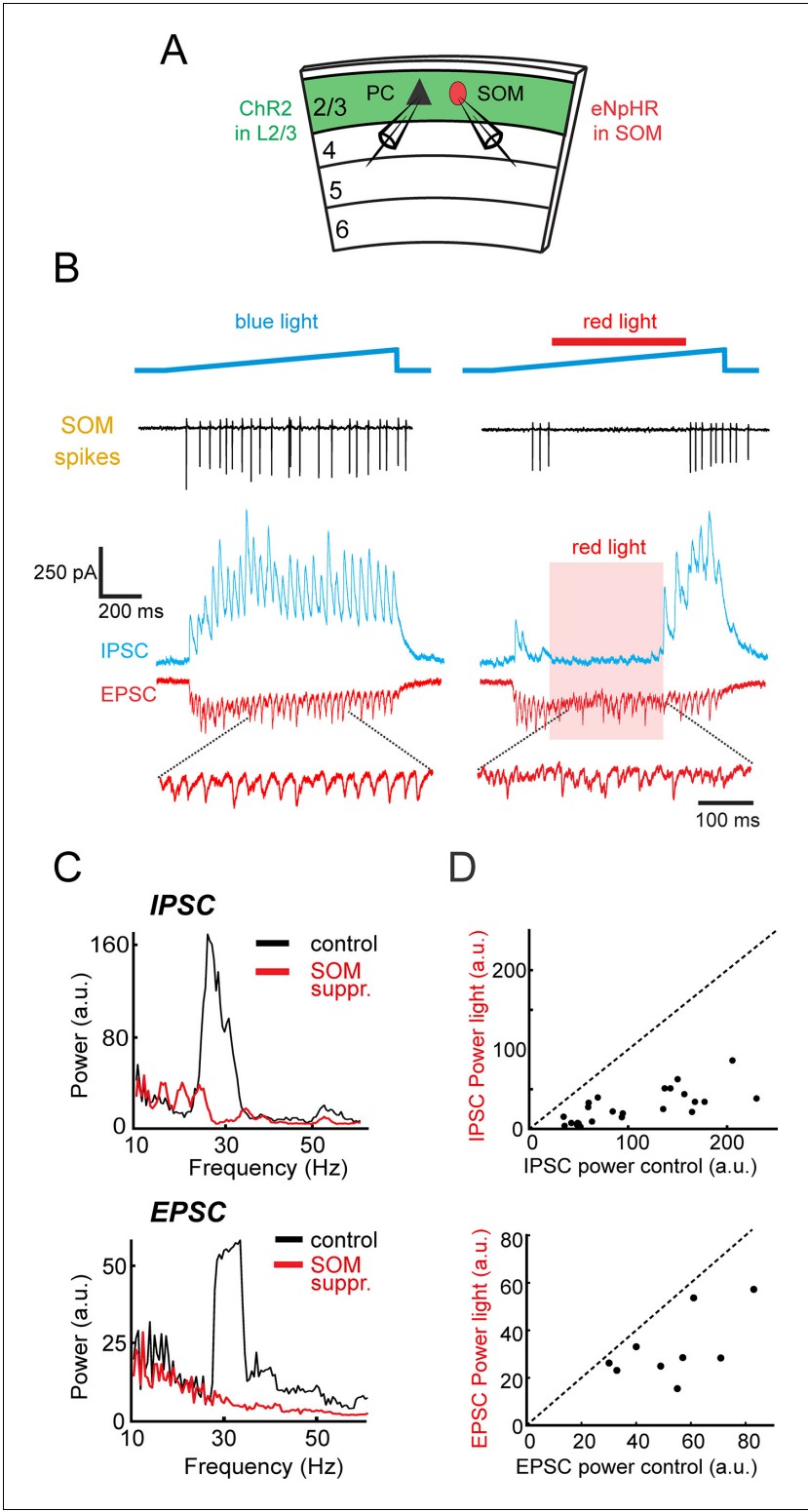

**Figure 3.** L2/3 gamma rhythms depend on SOM neuron activity. (**A**) Schematic of experiment. Whole-cell recordings are made from L2/3 Pyramidal cells (PC) in slices containing ChR2 in L2/3 and eNpHR3.0 in SOM cells. In some experiments a SOM cell was simultaneously recorded. (**B**) Left: loose-patch recording of spiking in an example eNpHR3.0 expressing-SOM cell (black, top trace) and the photo-induced excitatory (red) and inhibitory (blue) currents in a representative PC. Right: Same cells, but with the addition of red light (red bar) to suppress SOM cells. Bottom: expanded excitatory currents. (**C**) Power spectra of the recorded inhibition (top) and excitation

*Figure 3 continued on next page*

*Figure 3 continued*

(bottom) in the PC in B) under control conditions (black) and during photo-suppression of SOM cells (red). (D) Scatter plots of the peak gamma power of inhibition (top) and excitation (bottom) with and without photo-suppression of SOM cells (Inhibition: n = 24, $p<10^{-5}$, Wilcoxon signed rank test) (Excitation: n = 9, p<0.01, Wilcoxon signed rank test).

DOI: https://doi.org/10.7554/eLife.28569.006

The following figure supplements are available for figure 3:

**Figure supplement 1.** PV-neuron suppression does not significantly affect gamma oscillations.

DOI: https://doi.org/10.7554/eLife.28569.007

**Figure supplement 2.** SOM neurons mediate both initiation and maintenance of gamma oscillations.

DOI: https://doi.org/10.7554/eLife.28569.008

We, therefore, tested whether optogenetically suppressing SOM neurons at the induction of gamma oscillations would also abolish gamma rhythms. We found that suppressing SOM cells from before gamma induction also nearly completely abolished gamma rhythms, indicating that they are required even at the very earliest stages of gamma induction (*Figure 3—figure supplement 2F–H*). These results are consistent with direct recording of the spiking of SOM and PV cells during the time course of photo-induced gamma. We found that even at the early stages of gamma, SOM neuron activity still far exceeded that of PV neurons (*Figure 3—figure supplement 2A–E*). Although these data do not exclude the possibility that PV neurons could be necessary for oscillatory activity in response to other photo-stimulation protocols, they demonstrate that SOM neurons are necessary for both the initiation and maintenance of L2/3-induced gamma oscillations studied here.

## Lateral entrainment of pyramidal cells requires SOM activity

Whereas the preceding experiments establish that SOM neurons that are local to the oscillating ensemble are required for maintaining photo-induced gamma oscillations, the circuits that couple distal ensembles remain uncertain. Therefore, we tested whether SOM neurons are also involved in phase-locking spatially separated ensembles. We hypothesized that the horizontally projecting axons of L2/3 PCs might synchronize gamma rhythms across the retinotopic axis of the cortex by synapsing onto laterally located SOM cells, which then enforce phase-locking through their inhibitory output. To test this hypothesis, we optogenetically induced gamma activity in one region of the slice with a blue light patch, while inducing action-potentials in a ChR2-negative PC in different region of the slice using whole-cell somatic current injection (*Figure 4A,B* and *Figure 4—figure supplement 1*). We used current injection in the distal PC to evoke action potentials since the horizontal synaptic input from photo-stimulated L2/3 PCs tends to exert a strong net suppressive effect and does not evoke spiking on its own. The distance between the distal current-injected pyramidal cell and the border of the blue light was kept to approximately 325 µm, which maintained robust spike-oscillation entrainment (*Figure 4B–C*). The blue light region extended 600 µm laterally (950 µm maximum separation between distal PC and blue light region, see *Figure 4—figure supplement 1C*). Therefore, under these conditions, if suppressing SOM cells local to the current-injected pyramidal cell were to abolish gamma entrainment, it would support the notion that local SOM cell activity is required for both local and long-range gamma band synchronization.

In control conditions, the spiking of the current-injected neuron strongly phase-locked to the distally generated gamma oscillations (*Figure 4C*). In interleaved trials, we inactivated SOM cells specifically near the distal target PC with a patch of red light that did not overlap with the gamma oscillation initiation zone (see schematic, *Figure 4A*, and *Figure 4—figure supplement 1*). Optogenetic suppression of local SOM neurons with red light abolished phase-locking between PC-spikes and the distally generated gamma oscillation (*Figure 4C,D*, n = 8 pairs, PPC control: 0.120 ± 0.008; PPC red light: 0.004 ± 0.007, p=0.008, Wilcoxon signed-rank test. Data analyzed from 200 to 1000 ms post-stimulation onset.). Together, these data support the notion that the activity of SOM neurons outside of the gamma initiation zone is critical for phase-locking spatially separate oscillating ensembles across L2/3 of V1 in brain slices at least 325 µm apart. However, it should be noted that direct optogenetic suppression of SOM terminals, and partial suppression of SOM neurons between the blue and red light zones might also contribute (see *Figure 4—figure supplement 1F*)

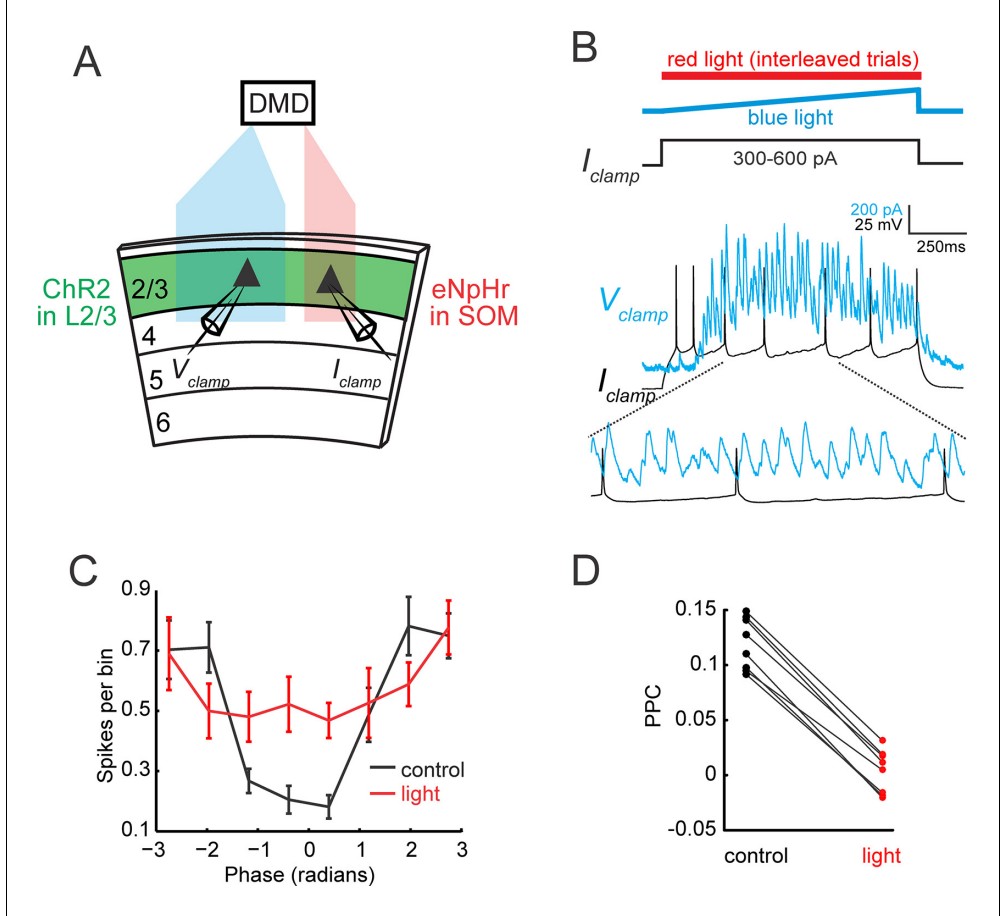

**Figure 4.** Gamma band synchronization across the retinotopic axis depends on local SOM neuron activity. (**A**) Schematic of experiment: One area of the slice is photo-stimulated with blue light to generate a local gamma oscillation, while the second distal site is simultaneously illuminated with red light to suppress local SOM activity focally (see *Figure 4—figure supplement 1C*). (**B**) Example traces from a simultaneously recorded pair of L2/3 PCs. The cell in the blue light zone is recorded in voltage clamp (blue, IPSC), while the cell in the red light zone is recorded in current clamp and stimulated with a current injection to evoke action potentials (gray). Current injections ranged between 300 and 600 pA. Bottom: Expanded trace showing spikes occurring at the toughs of inhibition. (**C**) Population phase histogram of the spikes of the current injected L2/3 PC in the red light zone under control conditions (blue light only) and during photo-suppression of local SOM cells relative to the phase of the gamma oscillation recorded in the second PC in the blue light zone. Error bars are s.e.m. (**D**) Scatter plot of the pairwise phase consistency of the L2/3 PC recorded in the red light zone relative to the oscillation recorded in the L2/3 PC in the blue light zone ($p < 10^{-7}$, n = 8 pairs, Wilcoxon signed-rank test).

DOI: https://doi.org/10.7554/eLife.28569.009

The following figure supplements are available for figure 4:

**Figure supplement 1.** Spatial resolution controls for DMD-based optogenetic suppression of SOM neurons.

DOI: https://doi.org/10.7554/eLife.28569.010

**Figure supplement 2.** Horizontal input to SOM neurons and PCs.

DOI: https://doi.org/10.7554/eLife.28569.011

In our first line of experiments (*Figure 1*), we observed significant coherence even for ensembles 625–850 μm apart, albeit less so compared to ensembles 275–400 μm apart. This long-range coherence, and its dependence on distance, could be explained by multiple mechanisms: long-range L2/3 excitatory axons to distal SOM neurons, mid-range L2/3 excitatory axons to SOM neurons located between the two ensembles and horizontal projections from these same SOM neurons, recurrently propagating loops of neural synchronization, or electrical coupling between SOM neurons that could facilitate synchronization across the retinotopic axis of V1. To begin to address these possibilities,

we mapped the range of excitatory input to SOM neurons from PCs by patching SOM neurons and stimulating increasingly more distant ensembles of L2/3 PCs (*Figure 4—figure supplement 2A*). We found that SOM neurons received significant, albeit attenuated excitatory input from L2/3 PCs as far away as 450 μm (*Figure 4—figure supplement 2B,C*, average maximum excitatory current: 120 ± 90 pA, 6 ± 2% of the maximum, n = 9). In current clamp, the same neurons reached an average maximal depolarization at 450 μm (calculated in in the final 100 ms of the photo-stimulus) of 7 ± 5 mV (n = 7), which was rarely enough to drive spiking on its own. At the same time, inhibitory currents in L2/3 PCs decayed to 4 ± 1% of their maximum at the same 450 μm distance (*Figure 4—figure supplement 2B,C*, maximum inhibitory current 20 ± 10 pA, n = 8).

When recording SOM neurons, we did not observe any clear spikelets or evidence of subthreshold electrical coupling that might mediate signal propagation between SOM neurons across the retinotopic access of the cortex (*Figure 4—figure supplement 2D–G*). However, it is possible that spikelets would be temporally filtered and therefore indistinguishable from EPSPs or EPSCs (*Hu and Agmon, 2015*), particularly against the large barrages of glutamatergic inputs SOM cells receive; thus the contribution of gap junctional coupling requires further exploration. When we examined the spatial spread of disynaptic inhibition onto PCs, inhibition exhibited a similar spatial decay as that of excitatory input to SOM neurons (*Figure 4—figure supplement 2B,C*). These data are consistent with a mechanism for long-range coherence involving the summed horizontal dendritic and axonal fields of pyramidal and SOM neurons.

## Discussion

Using patterned illumination optogenetics and multiple targeted intracellular recordings from various cortical neuron subtypes, this study establishes a novel circuit that is sufficient for synchronizing independently oscillating neuronal ensembles in the gamma band. Independent oscillating ensembles, in separate patches of V1, mutually connect to each other via disynaptic circuits composed of horizontally projecting excitatory axons of L2/3 neurons and SOM neurons.

These data support the conclusion that the gamma oscillations we observed here are generated locally by a PING (Pyramidal-Interneuron-Gamma) circuit motif (*Buzsáki and Wang, 2012*; *Tiesinga and Sejnowski, 2009*) between pyramidal neurons and cortical inhibitory neurons, with SOM neurons playing a critical role. Because the oscillation is induced by the selective photo-stimulation of excitatory neurons, and since SOM neurons largely lack recurrent synaptic connectivity in L2/3 (*Pfeffer et al., 2013*), an ING (Interneuron-Gamma) model appears unlikely. In vivo, visually induced gamma rhythms in mouse V1 are highly sensitive to SOM neuron suppression, whereas PV activity appears to be necessary for circuit stabilization (*Veit et al., 2017*). Taken together with previous work showing that L2/3 horizontal axons preferentially recruit SOM neurons (*Adesnik et al., 2012*), the data presented here support a model for phase-locking spatially separated ensembles across V1. Gamma rhythms, generated locally by a PING network in independent patches of V1, mutually connect to each other via circuits composed of horizontally projecting excitatory axons of L2/3 neurons and SOM neurons, which can extend the range of synchronization through their own horizontal projections. The recruited SOM inhibition synchronizes the two ensembles, even over relatively long distances. Although it remains to be further explored, since SOM interneurons are known to be gap junction-coupled (*Gibson et al., 1999*; *Hu and Agmon, 2015*), electrical connectivity among the SOM population might facilitate or reinforce gamma rhythm generation. We did not observe clear indications of spikelets in SOM neurons during horizontally propagating gamma oscillations that would support this hypothesis, although temporal filtering of spikelets could make distinguishing them from normal glutamatergic EPSPs challenging. Additionally, the impact of VIP neurons, which are particularly abundant in L2/3 and project strongly to SOM cells (*Pfeffer et al., 2013*), may also influence gamma entrainment by SOM cells, possibly in state-dependent manner (*Fu et al., 2014*).

Using circuit mapping approaches, we explored the mechanisms underlying our observation that oscillating ensembles exhibit significant coherence across long distances (>800 μm) across the retinotopic axis of V1. When we recorded from SOM cells and mapped their horizontal excitatory input from one side, by 450 μm excitatory input to SOM neurons was 6 ± 2%of its maximum value. Although in most cases this input was insufficient to drive any spikes on its own, it still led to significant depolarization of the membrane potential. These measurements probably underestimate the

true range of input to SOM neurons due to the sectioning of axons during brain slice preparation, but they do not support the notion that horizontal axons from the directly photo-stimulated L2/3 PCs are sufficient to explain the long-distance entrainment we observe beyond 450 µm. Our results most closely align with the model that SOM cells located between distant ensembles are driven by the sum of the input from the two sources, and synchronize the two ensembles through their own medium range projections. Interestingly, this architecture potentially resembles a phase-locked loop control system, with SOM neurons acting as phase comparators (*Ahissar, 1998*). For more closely spaced ensembles (<450 µm apart), horizontal input from each PC ensemble is sufficient to directly drive spiking of SOM cells local to the other ensemble.

In experiments using spatially restricted red light to inhibit SOM neurons near a phase-locked PC (*Figure 4*), it is possible that some of the reduction in phase-locking could be due to direct suppression of long-range SOM cell axons originating from the gamma oscillation initiation zone, or to direct somato-dendritic suppression of SOM cells located between the blue and red light zones. Synaptic terminal suppression by eNpHR3.0, though possible here, appears to be effective only early in an action potential train, at least in thalamic axons in which it was tested (*Mahn et al., 2016*). Since, during gamma rhythms, SOM cells fire prolonged spike trains up to 30 Hz, it seems unlikely that the effects we observed would be due to synaptic terminal suppression per se since illuminated axons would be able to maintain transmitter release after the first action potential. Alternatively, somato-dendritic hyperpolarization of SOM neurons located between the blue light and red light zone is plausible. While only a minority of SOM cells in L2/3 of V1 possess long-range horizontal axonal projections (*McGarry et al., 2010*), as suggested above, these few SOM neurons might be sufficient to help mediate the gamma band coherence.

While our data demonstrate the critical role SOM cells play in horizontally propagating gamma rhythms, in many circuits, both in vivo and in vitro, PV neurons also fire and synchronize to gamma rhythms, and in some cases, are necessary or sufficient to drive gamma rhythmicity (*Sohal et al., 2009*; *Cardin et al., 2009*). Several factors might account for the apparent contradiction with the previous literature. First, unlike these two previous studies where PV neurons were manipulated directly, we chose to directly activate excitatory neurons and allow the downstream circuit architecture to dictate the resulting network activity and dynamics. This suggests that while PV neurons or other mechanisms may be sufficient to drive gamma rhythmicity, the network architecture of L2/3 preferentially recruits SOM cells to fulfill this role. Moreover, recent results have shown that repeating the exact experiments of these two optogenetic studies with SOM cells could reproduce the same effects, demonstrating that SOM neuron activity is likewise sufficient to entrain gamma on its own (*Veit et al., 2017*)(*Chen et al., 2017*).

Second, gamma rhythms are phenomenologically diverse, exhibiting a range of peak frequencies, receptive fields, neuron subtype spike-time coupling, and other features (*Buzsáki and Wang, 2012*; *Colgin et al., 2009*; *Gieselmann and Thiele, 2008*; *Niell and Stryker, 2010*; *Ray and Maunsell, 2010*; *Saleem et al., 2017*; *Sohal et al., 2009*). This diversity strongly suggests diverse underlying mechanisms of oscillation generation and modulation across brain areas and network states. For this reason, care should be taken to compartmentalize conclusions on the underlying mechanisms of gamma generation based on differences in phenomenology and experimental approach.

Third, it is possible that in previous studies that observed tight coupling of PV (or 'fast-spiking' putative PV neurons) to ongoing gamma rhythms, the coupling is actually a result of gamma entrainment by SOM cells (which synapse strongly onto PV neurons) and not due to the PV neurons generating the gamma rhythm themselves. In mouse V1, PV neurons strongly couple to visually induced gamma, but optogenetically suppressing SOM, but not PV neurons, disrupts gamma rhythms in vivo (*Veit et al., 2017*), further suggesting this possibility.

Fourth, although our data demonstrate that SOM neurons are required for both local gamma generation and driving gamma-band coherence across the horizontal axis of the cortex, our data do not exclude a potential role for PV and other interneuron subtypes, such as VIP cells or deep-layer Martinotti cells. During gamma generation, PV neuron spikes, though sparse, are phase-locked to the gamma rhythm, and as a population may help reinforce the entrainment of local and/or distal excitatory neurons. This possibility is balanced with previous data showing that increasing stimulus sizes trend towards SOM neuron recruitment and away from PV neurons recruitment in vivo (*Adesnik et al., 2012*), which is in agreement with our findings in vitro, and our data showing that inhibition of distal SOM cells essentially abolishes distal entrainment to gamma rhythms. Moreover,

although the oscillations we observed were strongly dependent on horizontal connections across L2/3 only (*Figure 1G*), gamma rhythm generation could involve intracolumnar SOM cells in lower layers as well (*Kapfer et al., 2007*).

Horizontal circuits that engage SOM neurons are known to be important for contextual modulation in V1 (*Adesnik et al., 2012*), and recent data also show them to be critical for size-dependent visually induced gamma rhythms in mouse V1 (*Veit et al., 2017*), and in the olfactory bulb (*Lepousez et al., 2010*). Moreover, since SOM cells target PC dendrites, their inhibitory action during gamma oscillations could have significant impact on spike timing dependent plasticity in synapses located on distal dendrites. Taken together with our findings that SOM neurons are sufficient for gamma generation within L2/3 and drive gamma-band synchrony in distal neurons, SOM neurons are emerging as critical contributors to both gamma generation and long-range coherence in the cortex. Moreover, these findings suggest a new route to probe functional roles of gamma oscillations and gamma-band coherence in visual perception: direct manipulation of SOM neurons or their synaptic partners.

## Materials and methods

### Transgenic mice

All experiments were performed in accordance with the guidelines and regulations of the ACUC of the University of California, Berkeley and the IACUC of the University of California, San Diego. Both female and male mice were used. In vitro experiments were performed on animals aged 21–30 days old. Mice strains used in this study were wild-type ICR mice (Charles River), SOM-IRES-Cre (JAX stock 013044), PV-Cre (JAX stock 008069), and Rosa-LSL-tdTomato (JAX stock 007909). Mice were subjected to *in utero* electroporation at E15-E16 as previously described (*Adesnik et al., 2012*), which exclusively labels L2/3 pyramidal cells.

### Viral injection

Neonatal SOM-Cre mice (P1-3) were briefly cryo-anesthetized and placed in a head mold. Transcranial injection of ~45 nl of undiluted AAV9-DIO-Ef1a-eNpHR3.0-YFP (UPenn Vector Core) was performed using a Drummond Nanoject injector at three locations in V1. With respect to the lambda suture coordinates for V1 were 0.0 mm AP, 2.2 mm L and injection was 150–400 µm under the skull. For the experiments in *Figure 3—figure supplement 1*, the same virus was injected into juvenile PV-Cre mice (P14-P18) at 0.0 AP, 2.75 L, and 150–300 µm under the skull.

### In utero electroporation

Timed pregnant wild-type ICR mice were either purchased directly or timed-matings were set up between female ICR white mice (Charles River) and homozygous male SOM-IRES-CRE;Rosa-LSL-tdTomato or PV-CRE;Rosa-LSL-tdTomato mice. Pregnant mice at E15-16 were anaesthetized with 2.0% isoflurane, the abdomen was cleaned with 70% ethanol and swabbed with iodine, and a small vertical incision was made in the skin and abdominal wall and 8–12 embryos gently exposed. Each embryo was injected with 0.5–1 µl of DNA solution and 0.05% Fast Green dye. pCAG-ChR2-Venus plasmid DNA was mixed with pCAG-GFP for a total of 1–2 µg ChR2 DNA and 0.5–1 µg of fluorophore DNA per injection. Alternatively, pCAG-ChR2-mCherry without pCAG-GFP was used. We used a pressure-controlled beveled glass pipette (Drummond, Custom Microbeveller) for injection. After each injection, the embryos were moistened with saline and voltage steps via tweezertrodes (BTX, 5 mm round, platinum, BTX electroporator) were applied with the positive electrode placed over the visual cortex and the negative electrode placed under the head of the embryo. Voltage was 40 V for 5 pulses at 1 Hz, each pulse lasting 50 ms. The embryos were returned to the abdomen, which was sutured, followed by suturing of the skin. The procedure typically lasted under 30 min. On the day of birth, animals were screened for location and strength of transfection by trans-cranial epifluorescence under an Olympus MVX10 fluorescence stereoscope.

### In vitro recording

Mice were deeply anesthetized with isoflurane and quickly decapitated. 400 µm thick slices were cut on a microslicer (DTK-1000) and incubated at 34 degrees for 30–45 min, and then at room

temperature in sucrose cutting solution (in mM: NaCl, 83; KCl, 2.5; MgSO$_4$, 3.3; NaH$_2$PO$_4$, 1; NaHCO$_3$, 26.2; D-glucose, 22; sucrose, 72; and CaCl2, 0.5, bubbled with 95% O2% and 5% CO$_2$). A slice was transferred to a submerged chamber perfused with warmed (32 degree Celsius) ACSF (in mM: NaCl, 119; KCl, 2.5; NaH$_2$PO$_4$, 1.3; NaHCO$_3$, 26; D-glucose, 20; MgCl$_2$, 1.3; CaCl$_2$, 2.5; and mOsm, 305, bubbled with 95% O2% and 5% CO$_2$) and held down with nylon threads on a platinum harp. Excitatory and inhibitory currents were recorded in the voltage clamp mode with a cesium based internal solution (in mM: CsMeSO$_4$, 115; NaCl, 4; HEPES, 10; Na$_3$GTP, 0.3; MgATP, 4; EGTA, 0.3; QX-314-Cl, 2.5; BAPTA-5Cs, 10), and action potentials were recorded in a solution in which cesium was exchanged with potassium and QX-314 and BAPTA were omitted. Patch pipettes had resistances of 2–3 MOhm. Signals were amplified with two Multiclamp 700B amplifiers (Molecular Devices), filtered at 2 kHz and digitized via a National Instruments A/D card at 20 kHz. Custom software in Matlab (Mathworks) controlled all aspects of the experiment. Whole cell recordings were made from a L2/3 pyramidal shaped neurons. In some experiments, simultaneous recordings were made between two pyramidal cells or a pyramidal cell and a fluorescently labeled interneuron. Cells that exhibited direct photocurrents were discarded, except in *Figure 1—figure supplement 2*, where the spatial profile of photocurrents were measured. For whole-cell recordings, series resistance was less than 25 MOhm (uncompensated), and if this value changed in any cell by more than 20% during the course of an experiment, the cell was discarded. The light stimulus was always centered on layer 2/3 unless otherwise noted.

In *Figure 4*, current injection was used to bring the distal Pyramidal cell in current clamp to spike at a rate of ~10 Hz. Current injection amplitudes ranged from 300 to 600 pA, which corresponds to roughly 150–450 pA above typical L2/3 Pyramidal cell rheobase (~150 pA) (*Guan et al., 2007*; *Lefort et al., 2009*; *van der Velden et al., 2012*).

## Optogenetic stimulation in vitro with a DMD

For testing size-dependence and spatial coherence of gamma-oscillations (*Figure 1*), blue light was generated using a 1W 445 nm diode laser (Ultralasers) and routed via a liquid light guide into a CEL5500 digital micromirror device (DMD) (Digital Light Innovations, Austin, Texas). The projection from the DMD was then collimated and integrated into the light path of the microscope, before being focused onto the slice chamber using a 5 × objective lens (Olympus). The width of the blue light regions was 300 µm in the horizontal axis, and the borders of the two regions were never separated by less than 150 µm when varying the distance between the two recorded neurons. Cells were patched roughly in the middle of their respective regions. Activity was strongly biased, but not completely restricted to the illuminated regions, potentially due to neuronal processes extending into the illuminated regions as well as light scatter in the tissue (*Figure 1—figure supplement 2*, *Figure 4—figure supplement 1*).

For *Figure 2 and 3*, blue and red light was generated using a multicolor LED light engine (Lumencor Spectra X, Beaverton, Oregon) controlled by digital outputs (NI PCIe-6353), and was then routed via a liquid light guide before being focused onto the slice chamber using a 5 × objective lens (Olympus, Japan). To generate ramps of blue light, the signal was frequency modulated (0–5 kHz). Optogenetic photocurrents and spiking followed the integral of the light power with no correlation to the frequency of the rapidly pulsed light.

For *Figure 4*, blue light was generated using a 5W 445 nm laser diode (Nichia NUBM44) and red light was generated using a 700 mW 638 nm laser diode (Oclaro HL63193MG). The red and blue beams were expanded and combined before passing directly into the CEL5500 digital micromirror device (DMD), then focused onto the slice chamber using a 5 × objective lens (Olympus). The width of the blue light region was 600 µm in the horizontal axis, the width of the red light region was 350 µm, the separation between the borders of the red and blue regions was 150 µm (*Figure 4—figure supplement 1C*), and cells were patched close to the middle of their respective regions. In order to achieve two color spatial light modulation simultaneously with a single DMD, we temporally multiplexed the two color channels. The spatial patterns for the red and blue light alternated at 2.5 kHz, while we synchronized the triggering of the respective blue and red laser diodes to their corresponding light patterns (*Figure 4—figure supplement 1A,B*). A 2.5 kHz switching rate was sufficient to avoid any temporal cross talk between the two DMD patterns and the on-rate and off-rate of the laser diodes.

For all experiments using the DMD (*Figure 1 and 4*), the photo-stimulation patterns were calibrated and aligned to the slice chamber. Prior to photo-stimulation, infrared and epifluorescence images were captured using an IR-1000 CCD camera (DAGE-MTI, Michigan City, Indiana) and imported into MATLAB. These images were used define the borders for photo-stimulation. The DMD was used to pattern light into a rectangular region that was 600 µm long in the dorso-ventral axis, and of variable length in the horizontal axis.

In all experiments using ramps of light to induce oscillations, blue light intensity was ramped from zero to a final intensity that was determined to maximize the power of inhibitory gamma oscillations in the recorded pyramidal cell, which was titrated prior to making recordings. The determined final blue light intensity varied slightly between slices and animals (1.0–1.3 mW mm$^2$). Red light intensity for activation of eNpHR3.0 was 9.8 mW/mm$^2$ in experiments using wide-field illumination (*Figure 3*), and at 3.6 mW/mm$^2$ in experiments using DMD-based patterned illumination (*Figure 4*). For experiments in *Figure 3—figure supplement 1*, ChR2 was activated with a 1 mm multimode fiber-coupled blue LED mounted under the microscope objective, adjusted to illuminate all of V1. eNpHR3.0 was activated with a Hg$^+$ lamp long pass filtered (>600 nm), and gated by a electromagentic shutter (Uniblitz) through a 40x objective (Olympus).

## Data analysis

Analysis was performed in Matlab (Mathworks). Power spectra were computed using multi-tapered fourier estimation in Matlab with the Chronux package (http://chronux.org/)(*Mitra and Bokil, 2008*) using three tapers. Coherence was calculated as the magnitude-squared coherence using Welch's overlapped averaged periodogram method. To obtain instantaneous phase information to correlate to the spike times of interneurons, a Hilbert transform was computed on the time series of inhibitory currents. Spike-oscillation coherence was measured using the pairwise phase consistency metric (PPC) described in *Vinck et al. (2012)*. Synaptic charge during gamma activity was computed using trapezoidal integration of the voltage clamped currents at the corresponding reversal potentials for excitation (0 mV) and inhibition (−70 mV) after correction for the liquid junction potential (~7–10 mV) and baseline subtraction (mean of activity from t=-100 ms to t = 0 ms, relative to stimulus onset). For analysis in *Figure 4—figure supplement 2C*, current was integrated across the entire photo-stimulation period and normalized to the value at 0 µm. Membrane potential was computed by averaged the $V_m$ during this same period. Maximum currents and depolarizations were computed by integrating currents or averaging $V_m$ in the final 100 ms of the photo-stimulus.

## Histology

To prepare histological sections, animals were anesthetized with a combination of ketamine (100 mg/kg) and xylazine (10 mg/kg) and perfused with cold PBS followed by cold 4% PFA. Brains were dissected and post-fixed for 2 hr at 4°C, rinsed 3 × 15 min in PBS, and cryopreserved for 24 hr in 30% sucrose in PBS at 4°C. Brains were then sectioned using a frozen microtome at a thickness of 40 µm. Confocal images of visual cortex were obtained and cells were manually counted using FIJI (ImageJ). For *Figure 1—figure supplement 1*: A top, B, and C, 400 µm acute slices used in physiology experiment were imaged.

## Acknowledgements

The authors thank J Veit for extensive discussions, D Taylor and A Naka for technical assistance, and J Veit, B Sabatini, J Assad, and R Born for a critical reading of the manuscript. This work was supported by NEI grant R01EY023756-01 and the New York Stem Cell Foundation. HA is a New York Stem Cell - Robertson Investigator.

## Additional information

### Funding

| Funder | Grant reference number | Author |
| --- | --- | --- |
| National Eye Institute | R01EY023756-01 | Hillel Adesnik |

| New York Stem Cell Founda- | Hillel Adesnik |
|---|---|
| tion | |

The funders had no role in study design, data collection and interpretation, or the decision to submit the work for publication.

## Author contributions

Richard Hakim, Conceptualization, Resources, Supervision, Funding acquisition, Writing—original draft, Project administration, Writing—review and editing; Kiarash Shamardani, Investigation, Methodology; Hillel Adesnik, Resources, Methodology, Project administration

## Author ORCIDs

Richard Hakim (ID) https://orcid.org/0000-0002-6991-1801
Hillel Adesnik (ID) http://orcid.org/0000-0002-3796-8643

## Ethics

Animal experimentation: All experiments were performed in accordance with the guidelines and regulations of the ACUC of the University of California, Berkeley and the IACUC of the University of California, San Diego. Protocol # AUP-2014-10-6832

## Decision letter and Author response
Decision letter https://doi.org/10.7554/eLife.28569.014
Author response https://doi.org/10.7554/eLife.28569.015

# Additional files

## Supplementary files
• Transparent reporting form
DOI: https://doi.org/10.7554/eLife.28569.012

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
