## [Decision Letter]

Thank you for submitting your article "A neural circuit for long-range γ-band coherence in mouse visual cortex" for consideration by *eLife*. Your article has been reviewed by 3 peer reviewers, and the evaluation has been overseen by a Reviewing Editor and Timothy Behrens as the Senior Editor. The reviewers have opted to remain anonymous.

The reviewers have discussed the reviews with one another and the Reviewing Editor has drafted this decision to help you prepare a revised submission.

The study by Hakim and Adesnik examines the role of SOM-expressing interneurons in the synchronization of principal cell (PC) assemblies in the Layers 2/3 of the visual cortex slice preparation over large horizontal distances. All three reviewers judged the manuscript as important and interesting contribution to the field of neuroscience and as well performed. However, some concerns and questions were formulated from which the most important ones will be formulated in the following.

Previously, parvalbumin (PV)-expressing interneurons have been proposed to synchronize activity over large distances. The reviewers therefore ask for a discussion on the apparent dichotomy between the findings in this study on the synchronizing effect of SOM interneurons compared to PV cells. One of the reviewers asked whether due to the depressing nature of PV output synapses during repetitive activation and the facilitating characteristics of SOM interneuron outputs, γ oscillations generated at the onset of a strong sensory input may be PV mediated, while the sustained components later in the response could be SOM cell mediated. The reviewer therefore proposed to analyze γ expression using a wavelet method in sliding, 100 millisecond bins across the ramp stimulation period, and see if cell recruitment varies as a function of time period. Moreover, the authors need to be very clear in this manuscript over what period their analysis (e.g., of spiking rate, or phase locking) was conducted. The authors should weigh in on the question of why their results seem so different from prior studies in other brain areas. The authors conclude from their work that the oscillations are based on a PING mechanism. However, other mechanisms could also contribute such as gap junctional coupling between SOM cells and disynaptic inhibitory circuits involving VIP interneurons. The authors should mention these potentially involved mechanisms.

The study raised several questions related to the cellular mechanisms which may cause long-range synchronization which should be addressed in the Discussion of the manuscript: How do the SOM interneurons become synchronized to the far-away oscillation, and what are the limits on this? Are the local SOM interneurons receiving long-range excitatory input? Alternatively, do the SOM interneurons somehow transmit synchrony (e.g., through gap junctions) across the distance between the two sites? Or are sporadic excitatory neurons recruited at various points between the two sites? Do the SOM interneurons recruited at the site of the oscillation send long-range projections that inhibit the recorded pyramidal cell?

One of the reviewers was concerned regarding the precise location of the recorded PC in relation to the size of the applied light spot and intensity of light. Particularly for experiments in which a light spot was applied to one region and recorded a PC outside of that light spot, the distance between the edge of the light spot and the recorded PC was only ~200 µm. This suggests that inhibitory cells at the edge of the light spot might directly inhibit / entrain the recorded PC which does not qualify this as "long-distance" synchronization. To address this issue, the authors should record from the PC as a function of distance, preferably much farther from the edge of the light spot. A further important issue is that for experiments in which the authors looked at synchronization between two light spots, it would good to record from cells in between the two spots to examine how the two spots are communicating with each other. Since the main aim of the work was to examine the neuronal network underlying long-range synchronization of network oscillations, the experiments should be repeated as a function of distance.

Finally, the authors should quantify / characterize the frequency of the induced oscillations and mention this in the Abstract.

*Reviewer #1:*

This study examines the role of SOM interneurons in the synchronization of PC assemblies in Layer 2/3 of the visual cortex slice preparation over large horizontal distances. The authors provide evidence that long-range synchronization of γ activity patterns is driven by PCs projecting laterally and recruit local SOM cells (PING model). The study is overall well done. My main criticism relates to the question whether PV interneurons could support this long-range synchrony. In their previous work (Veit et al., 2017; Nat Neurosci) the authors tested the role of SOM cells in long-range synchrony in vivo and showed that optogenetic silencing of SOM cells reduces cross-correlation in Γ activity between both the local and more distant cortical sites. The effect of PV cells on long-range γ synchronization, however, was not tested. Moreover, is the role of SOM cells in long-range synchronization indeed layer-dependent? SOM cells are also located in other cortical layers but PCs do not project over large distances to other cortical regions (at least not as in layer II/III), similar results may not emerge in layer V. Please provide evidences for this assumption. Overall the manuscript is nice but it has the 'taste' of a secondary study of the Veit et al., 2017 paper.

*Reviewer #2:*

These experiments are well-conducted and will make an important contribution to the literature. They were cleanly designed to make the conceptual advancement argued for. There are several points that need to be addressed before publication:

I would recommend that the authors discuss the apparent dichotomy between their finding that SOM plays a critical role in γ generation, and previous literature attributing this role mainly to PV cells.

Could the difference be related to the integration window over which γ is measured? SOM neurons are known to show facilitating responses after a period of prior input, usually lasting tens to hundreds of milliseconds (but, certainly lasting at least a few γ cycles), whereas PYR → PV neurons can depress on the same time scale. If the authors only analyzed γ in the early phase of the response, is the disparity as great? I'm obviously guessing that early γ (soon after onset, say for the first 4 cycles) is PV mediated, while late γ (analyzed substantially after sensory/ChR2 onset, say 1+ seconds) is 'taken over' by the recruitment of SOM. If so, this temporal hand-off/evolution has interesting implications, and is a major step towards reconciling the prior relatively overwhelming literature showing γ mediated by FS-PV. In this vein, ramp-up stimulation to PCs is ideal for generating sparse activity across a pool of PCs initially that will drive PV suppression, and then as input gets stronger, optimal SOM recruitment. This comment is obviously (I hope!) not a critique of ramps, they likely replicate some kind of natural dynamic in the brain, but it does also help explain some of the differences in the literature and these current findings. Specifically, the γ generated at the onset of a strong sensory input – either bottom up salient or due to attention-mediated inhibition – may be PV mediated, while the sustained components later in the response (reflecting working memory, enhanced discriminative capability, etc.) could be SOMian.

• Recommendation I: Analyze γ expression using a wavelet method in sliding, 100 millisecond bins across the ramp stimulation period, and see if cell recruitment varies as a function of time period. I would also recommend inclusion here of such analysis applied to the recent Veit data generated with real sensory stimuli.

• Recommendation II: The authors need to be very clear in this manuscript over what period their analysis (e.g., of spiking rate, or phase locking) is conducted.

As a final, somewhat conceptual point: While the 'firewall' the authors place around their data as being about visual neocortex is fair, it implies that the differences from many prior papers in hippocampus and SI results from an area-of-the-brain issue, not an evolution across seconds issue (as raised above) or any of many other possibilities. As such, they should weigh in on the specific question of why their results seem so different from prior studies. Again, my guess is time frame, making interesting implications for function, but whatever their rumination, they should state it clearly.

*Reviewer #3:*

The subject is interesting, the experiments well done, and the slice preparation appropriate for answering detailed mechanistic questions. The main potential novelty here is the ability to elucidate the basic mechanisms of the oscillation beyond what this lab has already done in vivo, and to work out the mechanisms of long-range synchronization. With regard to working out mechanistic details of the oscillation, the main contribution is to observe the inhibitory and excitatory synaptic currents in pyramidal, SOM, and PV interneurons. The authors conclude that the oscillation reflects a PING mechanism; this seems likely although other mechanisms that could be at play here – specifically gap junctional coupling between SOM cells and disynaptic inhibitory circuits involving VIP interneurons. I'm not sure exactly what the authors could easily do to sort out these issues, but they are slightly tangential to the main point, so I think the authors could simply mention them.

Regarding the role of SOM interneurons in long-range synchrony, the authors conclude that long-range synchronization is driven by SOM interneurons. It is certainly true that if you induce an oscillation in one region of the slice, while recording from (and injecting depolarizing current into) a pyramidal cell some distance away, then when you inhibit SOM interneurons in the vicinity of the recorded pyramidal cell, its synchrony to the induced oscillation breaks down. Of course this sort of has to be the case – the L2/3 cells strongly activate SOM cells, which inhibit PV cells – most pyramidal cells seem to receive mainly inhibitory input, so if you disrupt the major source of inhibitory input to this cell, then it is effectively decoupled from the oscillation. The real question in my mind, is how do the SOM interneurons become synchronized to the far-away oscillation, and what are the limits on this? I can imagine many possibilities: Are the local SOM interneurons receiving long-range excitatory input? Alternatively, do the SOM interneurons somehow transmit synchrony (e.g., through gap junctions) across the distance between the two sites? Or are sporadic excitatory neurons recruited at various points between the two sites? Do the SOM interneurons recruited at the site of the oscillation send long-range projections that inhibit the recorded pyramidal cell? The authors dismiss the possibility that optogenetic inhibition affects terminals here, but I think we don't really know whether this is happening – the efficacy of such optogenetic inhibition on terminals likely depends on the details of cell type and preparation.

Furthermore, for these experiments, the Materials and methods specify that the width of the blue light region was 600 um, the width of the red light region was 350 um, there was a 150 μm gap between these two regions, and they patched cells in the middle of each region. This is quite different than the impression conveyed by the schematic in Figure 4 (and Figure 4—figure supplement 1) and I would recommend adjusting these figures to be more reflective of the actual experimental condition. Based on this, the distance between the patched cell and the edge of the blue-light induced oscillation is at most ~325 μm – likely less given that Figure 1 seems to suggest that oscillation-induced pyramidal cell spiking extends ~100 μm past the border of the blue light region. If the distance between the pyramidal cells directly excited by the local oscillation and the recorded "distant" pyramidal cell is only ~225 um, then I'm not sure this really qualifies as "long-distance" synchronization and I'm not sure the mechanisms underlying this experiment are necessarily the same ones that mediate the synchronization of oscillations induced by light delivery to two different, more widely regions of the slice. Notably, the width of the blue light region for this experiment was 600 μm whereas for the other experiment (two light patches to induce oscillations at two locations) it was 300 um. This raises the question of whether synchronization in this experiment vs. during stimulation of two different more widely separated regions are really the same phenomenon. In particular, when the patch of blue light is larger, the distance between the edge of the induced oscillation and the distant recorded cell will be much smaller than the distance between the center of the stimulated region and the "distant" recorded cell.

The title and Abstract suggest that the main motivation for this study is identifying the neural circuit that mediates long-range synchronization of oscillations. This is a very interesting topic that is certainly worthy of publication in *eLife* and the methods used are appropriate. Based on the issues raised above, I'm not sure the authors actually reveal this mechanism. That being said, I am enthusiastic about the direction of this work and think this concern could be addressed by repeating these experiments as a function of distance, and possibly making recordings from pyramidal cells and/or SOM interneurons in the intervening region.

Finally, the authors should quantify/characterize the frequency of the induced oscillations and be specific about this in the Abstract since "γ" oscillations mean different things to different people.

[Editors' note: further revisions were requested prior to acceptance, as described below.]

Thank you for resubmitting your work entitled "A neural circuit for γ-band coherence across the retinotopic map in mouse visual cortex" for further consideration at *eLife*. Your revised article has been favorably evaluated by Timothy Behrens (Senior Editor), a Reviewing Editor, and 1 reviewer.

The manuscript has been improved but there are some remaining issues that need to be addressed before acceptance, as outlined below:

The reviewers fully support the study and find that the phenomena outlined here are very interesting, important, and novel. However, the mechanistic understanding remains incomplete. Therefore additional experiments are required. The reviewer asks for one set of important experiments: record from SOM cells and pyramidal cells at various distances from a single patch of blue light / ChR2 excitation, and evaluate whether these cells receive long-distance EPSCs, long-distance IPSCs, and/or currents that appear to be gap-junction mediated. If SOM interneurons 400-500 µm away from the blue light patch receive EPSPs but do not show evidence of gap-junction mediated currents (e.g. spikelets) and there are not IPSPs in pyramidal neurons at a similar distance, then this would provide definitive evidence in favor of the author's proposed mechanism.

*Reviewer #3:*

I completely agree with the authors that:

1) The role of SOM interneurons in γ-frequency synchronization has long been underappreciated by the field;

2) SOM cells are essential for synchronization in the second experiment (shown in Figure 4).

The two key questions, which in my mind remain unanswered, are:

1) Does the same mechanism support long-range synchronization in the first experiment (Figure 1) and shorter-range synchronization in the second experiment (Figure 4)?

2) In the second experiment, what is the source of SOM-mediated synchronization – is it gap-junction coupling, medium-range SOM projections, or medium range excitatory input to local SOM interneurons?

Specifically, with regard to #2: the authors argue that gap-junction coupling is not sufficient to extend the range of synaptically evoked inhibition. That may be true, but it doesn't mean that SOM interneurons don't receive subthreshold rhythmic gap junction mediated currents that elicit synchronization. The authors argue against the possibility of medium-range SOM projections, by arguing that eNpHR would not be effective for silencing terminals, based on published data from Mahn et al. I would point out that the situation here is quite different from Mahn et al. – the distance between the edge of the red and blue light patches is only 150 um. Given that the red light seems to inhibit cells ~100 μm away (Figure 4), and that the oscillation extends some distance beyond the edge of the blue light, it seems likely that the red light will directly suppress the activity of some SOM cells involved in the oscillation (not just inhibit their terminals).

So while the phenomena outlined here are very interesting, important, and novel, the mechanistic understanding remains incomplete. I am sympathetic to the experimental constraints noted by the authors. It may not be possible to repeat the second experiment using a smaller blue light patch/larger gap. However, given that this is a relatively short paper that builds on the author's previous work, my opinion is that there should be some test of how synchrony is transmitted by SOM cells. This could be quite minimal – I think simply recording from SOM cells and pyramidal cells at various distances from a single patch of blue light / ChR2 excitation, and evaluating whether these cells receive long-distance EPSCs, long-distance IPSCs, and/or currents that appear to be gap-junction mediated, would be very informative. I.e., if SOM interneurons 400-500 μm away from the blue light patch receive EPSPs but do not show evidence of gap-junction mediated currents (e.g., spikelets), and there are not IPSPs in pyramidal neurons at a similar distance, then this would provide definitive evidence in favor of the author's proposed mechanism.

The long-range transmission of synchrony (potentially) by SOM neurons is the key novel finding that differentiates this from the author's previous work, which is why I think this kind of additional evidence is important. If the authors were to find, for example, contrary to the preceding, that SOM cells at the border of the illuminated region (e.g., 50 μm beyond the edge of the blue light) provide inhibition to pyramidal cells ~200-300 μm away (very plausible given Figure 4 of Fino and Yuste), and SOM interneurons >300 μm away from the illuminated region do not receive significant EPSPs, then this would call into question the authors' interpretation.

---

## [Author Response]

Reviewer #1:

This study examines the role of SOM interneurons in the synchronization of PC assemblies in Layer 2/3 of the visual cortex slice preparation over large horizontal distances. The authors provide evidence that long-range synchronization of γ activity patterns is driven by PCs projecting laterally and recruit local SOM cells (PING model). The study is overall well done. My main criticism relates to the question whether PV interneurons could support this long-range synchrony. In their previous work (Veit et al., 2017; Nat Neurosci) the authors tested the role of SOM cells in long-range synchrony in vivo and showed that optogenetic silencing of SOM cells reduces cross-correlation in Γ activity between both the local and more distant cortical sites. The effect of PV cells on long-range γ synchronization, however, was not tested. Moreover, is the role of SOM cells in long-range synchronization indeed layer-dependent? SOM cells are also located in other cortical layers but PCs do not project over large distances to other cortical regions (at least not as in layer II/III), similar results may not emerge in layer V. Please provide evidences for this assumption. Overall the manuscript is nice but it has the 'taste' of a secondary study of the Veit et al., 2017 paper.

While we agree that a role for PV neurons cannot be entirely excluded in long range synchronization in our slices, and have added this to the Discussion, there are several pieces of data that make this possibility significantly less likely. First, when we optogenetically inactivated SOM neurons and looked at distal synchronization, the phase locking of the target pyramidal cells was very strongly reduced, and in most neurons essentially abolished, implying that SOM neuron activity alone is sufficient for the γ rhythmicity (Figure 4). Second, when we generate large, coherent γ oscillations across the entire slice with broad illumination, PV neurons are only very weakly recruited, while SOM neurons are very strongly recruited (Figure 2). If PV cells were critical for long range synchrony, one would expect PV neurons to be more strongly recruited. These points have been added to the Discussion as well. Third, under these same conditions where γ oscillations are coherent across nearly all of V1 in the brain slice (Figure 1), suppressing PV neurons has no significant effect on γ. Under all these conditions where long-range synchronization is very prominent, either PV neurons are inactive, not necessary, or SOM neurons account for nearly all of the observed coherence or γ power.

With respect to layer-specific recruitment of SOM cells in long-range γ entrainment: This is an important and fascinating question that we address more thoroughly in the Discussion, but requires further study in the future. In this work, we chose to constrain our focus to L2/3-generated γ oscillations for simplicity and clarity, and while our results point to critical roles of L2/3 SOM cells in the oscillations that we study, we do not exclude a role for SOM cells in other layers, or that oscillations generated in other layers may depend on other inhibitory neuron subtypes. One line of evidence strongly suggesting that L2/3 SOM cells, and not SOM cells in other layers (particularly L5 Martinotti cells), are mediating the long-range coherence is the experiment in Figure 1, which shows the necessity of lateral connections in *only* L2/3 on long-range coherence. However, Kapfer et al. (2007) show that recurrent connections between L2/3 PCs and L5 Martinotti cells, suggesting that local rhythm generation may involve deeper layer SOM cells as well.

Reviewer #2:

These experiments are well-conducted and will make an important contribution to the literature. They were cleanly designed to make the conceptual advancement argued for. There are several points that need to be addressed before publication:I would recommend that the authors discuss the apparent dichotomy between their finding that SOM plays a critical role in γ generation, and previous literature attributing this role mainly to PV cells.

We have substantially expanded on this point in the Discussion. It takes very minimal circuitry to generate γ rhythms (Sohal et al. 2009; Veit et al. 2017 (Figure 4)), and neural circuits in the cortex, hippocampus, and other areas are composed of a sufficiently large number of distinct cell types to easily support different γ generating mechanisms in different layers, subnetworks, or conditions. We now emphasize that in many circuits, both in vivo and in vitro, PV neurons are clearly shown to fire and synchronize to γ rhythms. However, we also point out that entrainment to γ and a role in its generation are potentially distinct. For instance, as Veit et al. recently demonstrated, PV neurons in the mouse visual cortex lock well to visually induced γ, but this rhythm is abolished by suppressing SOM and not PV neurons. This result implies that in studies where PV neurons locked to γ, the underlying oscillation might still be dependent on SOM and not PV neurons. Distinguishing between these possibilities requires causal manipulations, such as optogenetic suppression of SOM cells. However, in a few cases, causal manipulations of PV neurons have demonstrated that they are either necessary or sufficient for γ entrainment, such as in the barrel cortex (Cardin et al.) or in the prefrontal cortex (Sohal et al.). Our conclusion is that long-range synchronization in L2/3 of the visual cortex (at least in mice) is dependent more intimately on SOM neurons, but that γ rhythms in other brain areas, or even other layers may instead depend on PV neurons or alternative circuits. Since long-range synchronization in superficial V1 is one of the best studied models for γ rhythms, we consider our study impactful, whether or not the same mechanism may hold for the various γ rhythms observed in the hippocampus or prefrontal cortex.

Could the difference be related to the integration window over which γ is measured? SOM neurons are known to show facilitating responses after a period of prior input, usually lasting tens to hundreds of milliseconds (but, certainly lasting at least a few γ cycles), whereas PYR → PV neurons can depress on the same time scale. If the authors only analyzed γ in the early phase of the response, is the disparity as great? I'm obviously guessing that early γ (soon after onset, say for the first 4 cycles) is PV mediated, while late γ (analyzed substantially after sensory/ChR2 onset, say 1+ seconds) is 'taken over' by the recruitment of SOM. If so, this temporal hand-off/evolution has interesting implications, and is a major step towards reconciling the prior relatively overwhelming literature showing γ mediated by FS-PV. In this vein, ramp-up stimulation to PCs is ideal for generating sparse activity across a pool of PCs initially that will drive PV suppression, and then as input gets stronger, optimal SOM recruitment. This comment is obviously (I hope!) not a critique of ramps, they likely replicate some kind of natural dynamic in the brain, but it does also help explain some of the differences in the literature and these current findings. Specifically, the γ generated at the onset of a strong sensory input – either bottom up salient or due to attention-mediated inhibition – may be PV mediated, while the sustained components later in the response (reflecting working memory, enhanced discriminative capability, etc.) could be SOMian.• Recommendation I: Analyze γ expression using a wavelet method in sliding, 100 millisecond bins across the ramp stimulation period, and see if cell recruitment varies as a function of time period. I would also recommend inclusion here of such analysis applied to the recent Veit data generated with real sensory stimuli.

The reviewer proposes an intriguing hypothesis that we have now addressed through additional experiments and analyses. A temporal ‘hand-off’ in γ mediation from PV to SOM cells would indeed be revealing. To address this, first, we performed the analysis suggested by the reviewer (Figure 3—figure supplement 2) by aligning SOM and PV spike times to the onset of IPSCs. Alignment to the onset of IPSCs provided a more consistent and interpretable metric than alignment to the onset time of γ power using wavelet analysis, though a similar aggregate result emerges with this method. We find that PV spiking is sparse during all epochs of γ and reaches peak activity after SOM cells do (300-400ms for PV, 100-200ms for SOM). Second, we repeated the experiment in Figure 3, but initiated SOM suppression prior to the onset of inducing γ. We found that γ power remains abolished (Figure 3—figure supplement 2). These experiments demonstrate that SOM neuron activity is required for both the early and ongoing phase of layer 2/3 γ oscillations and that PV neurons are neither necessary nor specifically recruited for initiation or maintenance under our conditions. This said, we agree (and now add to the Discussion) that different temporal patterns of photo-stimulation (other than ramps) could potentially lead to conditions where PV neurons should fire at the onset of the response (as they do with a brief pulse of light – see Figure 3—figure supplement 1).

Consistent with well-known results from the hippocampus and Layer 4 of the barrel cortex, L2/3 PV neurons clearly respond well to brief, strong increases in afferent input, while SOM cells responds better to more sustained input. Thus while SOM cells appear sufficient in our conditions to initiate and maintain γ oscillations without much input from PV neurons, under ‘natural conditions’ – such as time varying input during natural vision, we would expect that PV neurons are periodically recruited, and their inhibitory output may be involved in phase modulating the γ rhythm on a cycle by cycle basis. This remains to be explored in the future. In Veit et al., 2017, we also showed that photo-suppression of SOM cells from before visual stimulus onset likewise impaired γ induction, demonstrating that SOM cells are involved even at the very earliest stages of γ entrainment. Similar photo-suppression of PV cells did not have the same effect, although we must cautiously interpret this experiment since it was not possible to suppress PV cells very strongly in awake mice without inducing epileptic events. Hence, in that paper we conclude that both SOM and PV cells are required for γ oscillations, but that PV cells’ role in this γ is primarily for network stabilization, while the SOM cells are more intimately involved in γ entrainment.

• Recommendation II: The authors need to be very clear in this manuscript over what period their analysis (e.g., of spiking rate, or phase locking) is conducted.

Additional text has been added to clearly define the time period over which our analyses are conducted. In experiments where interneuron subtypes were inhibited during γ (Figure 3, Figure 3—figure supplement 1, Figure 3—figure supplement 2, Figure 4), analysis was generally conducted during epochs where the red light was on or would have been on in complementary control trials: 250ms-750ms for Figure 3, Figure 3—figure supplement 1; 0-1000ms for Figure 3—figure supplement 2; and in Figure 4 we analyzed from 200-1000ms so as to ignore spikes that occurred prior to the emergence of steady oscillations and after the adapting period of regular-spiking L2/3 Pyramidal cells. In Figure 4—figure supplement 1, we analyzed during the red light epoch in control and red light trials (250-750 ms). For all other experiments (Figure 1, Figure 1—figure supplement 2, Figure 2) the entire stimulation period (0-1000ms) was used for analysis.

As a final, somewhat conceptual point: While the 'firewall' the authors place around their data as being about visual neocortex is fair, it implies that the differences from many prior papers in hippocampus and SI results from an area-of-the-brain issue, not an evolution across seconds issue (as raised above) or any of many other possibilities. As such, they should weigh in on the specific question of why their results seem so different from prior studies. Again, my guess is time frame, making interesting implications for function, but whatever their rumination, they should state it clearly.

We agree this is a critical point for discussion. As we stated above, we have substantially amended the Discussion and address the issue of temporal dynamics. It is true that many studies have found that PV or fast spiking (presumed PV neurons) typically lock to ongoing γ, and in a small number of studies, have been causally related to γ. However, we emphasize that locking to γ proves neither causality nor sufficiency, as PV neurons receive powerful inhibition from SOM neurons and can be entrained by SOM cells themselves, as we recently showed during visually induced γ rhythms in mouse V1 (Veit et al., 2017). Thus, we suspect that if this question were reexamined more broadly, a subset of the instances of γ in other brain circuits might also be found to depend on SOM neurons.

Reviewer #3:

The subject is interesting, the experiments well done, and the slice preparation appropriate for answering detailed mechanistic questions. The main potential novelty here is the ability to elucidate the basic mechanisms of the oscillation beyond what this lab has already done in vivo, and to work out the mechanisms of long-range synchronization. With regard to working out mechanistic details of the oscillation, the main contribution is to observe the inhibitory and excitatory synaptic currents in pyramidal, SOM, and PV interneurons. The authors conclude that the oscillation reflects a PING mechanism; this seems likely although other mechanisms that could be at play here – specifically gap junctional coupling between SOM cells and disynaptic inhibitory circuits involving VIP interneurons. I'm not sure exactly what the authors could easily do to sort out these issues, but they are slightly tangential to the main point, so I think the authors could simply mention them.

We have clarified that our statement that a PING mechanism likely underlies γ generation is a hypothesis that is suggested but not proven by our data, and as such, additional discussion on other plausible mechanisms, such as the role of electrical connections has been included.

Regarding the role of SOM interneurons in long-range synchrony, the authors conclude that long-range synchronization is driven by SOM interneurons. It is certainly true that if you induce an oscillation in one region of the slice, while recording from (and injecting depolarizing current into) a pyramidal cell some distance away, then when you inhibit SOM interneurons in the vicinity of the recorded pyramidal cell, its synchrony to the induced oscillation breaks down. Of course this sort of has to be the case – the L2/3 cells strongly activate SOM cells, which inhibit PV cells – most pyramidal cells seem to receive mainly inhibitory input, so if you disrupt the major source of inhibitory input to this cell, then it is effectively decoupled from the oscillation. The real question in my mind, is how do the SOM interneurons become synchronized to the far-away oscillation, and what are the limits on this? I can imagine many possibilities: Are the local SOM interneurons receiving long-range excitatory input? Alternatively, do the SOM interneurons somehow transmit synchrony (e.g., through gap junctions) across the distance between the two sites? Or are sporadic excitatory neurons recruited at various points between the two sites? Do the SOM interneurons recruited at the site of the oscillation send long-range projections that inhibit the recorded pyramidal cell? The authors dismiss the possibility that optogenetic inhibition affects terminals here, but I think we don't really know whether this is happening – the efficacy of such optogenetic inhibition on terminals likely depends on the details of cell type and preparation.

The reviewer brings up several important points that we address here and by modifying the Discussion. First, we agree with the reviewer’s intuition that the distal entrainment of spike-timing to fast oscillations by SOM neurons could potentially be predicted by what we know of cortical circuitry; however this prediction has not been experimentally demonstrated and has not been explicated as a prediction by many leaders in the field (Traub et al., 1996c, Bartos et al., 2007, Buzsaki and Wang, 2012). Therefore, we feel that demonstrating this result experimentally has important value.

Second, SOM neurons do receive significant long range excitatory input, as shown by experiments in Adesnik et al., 2012. We believe this long range excitatory input to SOM cells from L2/3 pyramidal cells is at the heart of the mechanism for γ synchronization under study here.

With respect to the possibility that SOM cells transmit synchrony via gap junctions: This is an intriguing idea that has some support both at the single cell level (Hu and Agmon, 2015) and network level (Connors, 2017). However, if we compare the spatial fall-off of pair-wise connectivity from SOMs onto PCs (Fino and Yuste, 2011) vs. the spatial fall-off of IPSC amplitudes recorded in PCs by stimulating populations of SOM cells at different distances (Kato et al., 2017), we see essentially identical spatial fall-off curves. This suggests that, at a network level, in layer 2/3 of cortical slices, SOM cells do not receive a significant boost to their spatial propagation due to gap-junctions, although this could be tested experimental in the future with appropriate methods to block or delete gap junctions in SOM cells.

With respect to the possibility that sporadic excitatory neurons are recruited at various points between two distantly-separated ensembles: In agreement with previous work demonstrating that layer 2/3 Pyramidal cells located outside the blue light stimulation receive strong lateral inhibition mediated by SOM cells and are net suppressed (Adesnik et al., 2010, 2012), we never found pyramidal cells outside the blue light illumination zone that were synaptically driven to spike (Figure 1—figure supplement 2). In the experiment in Figure 4, all recorded Pyramidal cells outside the blue light region required current injection (~300-600 pA, roughly 100-400 pA above typical rheobase (Guan et al., 2007; van der Velden et al., 2012; Lefort et al., 2008)) in order to induce spiking. In other words, our data indicate that the only L2/3 excitatory neurons that are spiking in the slice are those that are in the blue light regions, implying that those in between are not firing due to synaptic activation. This essentially rules out the possibility that the synchronization is propagated through multisynaptic loops across the slice.

As to whether SOM cells send long range projections, McGarry et al., 2010 and Fino and Yuste, 2011 found that a subset of SOM neurons do exhibit axons that ascend towards layer 1, make a 90 degree turn, and then project horizontally from anywhere between 100 and 400 hundred microns. Therefore, the reviewer is correct that if our red light stimulation of eNpHR3.0 did suppress synaptic release from SOM terminals, a possible interpretation is that these horizontally-projecting axons were critical. Definitively testing whether this was the case under our conditions has proven to be very tricky (experiments with co-expression of ChR2 and eNpHR3.0 in our lab have always ended up inconclusive). Nevertheless, the definitive study on this methodological problem, so far, is Mahn et al., Nature Neuroscience 2016. Figure 2 of that paper shows that illumination of eNpHR3.0 expressing terminals suppresses the response to the first spike in a two spike sequence, but *potentiates* the second. In other words, optogenetic suppression of terminals appears only to reduce the probability of synaptic release, but not suppress transmitter release under conditions of sustained presynaptic activity (as studied here, and as is typical in most in vivo scenarios). While this study did not examine SOM cells’ terminals specifically, we surmise that the same might be true, and that our red light illumination would only have very transiently suppressed GABA release from SOM cell axons. This suggests that axon terminals were not specifically inhibited in our experiment, and those horizontally projecting SOM cells maintained their output onto the recorded Pyramidal cell with and without red light. Please also note that under Mahn’s conditions the *average* suppression was only ~20% on the first synaptic (Figure 2). The dramatic example they show in Figure 2 is clearly not representative. Thus we conclude that terminal suppression with eNpHR3.0 is actually quite weak and largely ineffective during sustained activity. Nevertheless, we cannot yet rule this contingency out with certainty, and we bring this up as an important issue in the Discussion.

Furthermore, for these experiments, the Materials and methods specify that the width of the blue light region was 600 um, the width of the red light region was 350 um, there was a 150 μm gap between these two regions, and they patched cells in the middle of each region. This is quite different than the impression conveyed by the schematic in Figure 4 (and Figure 4—figure supplement 1) and I would recommend adjusting these figures to be more reflective of the actual experimental condition. Based on this, the distance between the patched cell and the edge of the blue-light induced oscillation is at most ~325 μm – likely less given that Figure 1 seems to suggest that oscillation-induced pyramidal cell spiking extends ~100 μm past the border of the blue light region. If the distance between the pyramidal cells directly excited by the local oscillation and the recorded "distant" pyramidal cell is only ~225 um, then I'm not sure this really qualifies as "long-distance" synchronization and I'm not sure the mechanisms underlying this experiment are necessarily the same ones that mediate the synchronization of oscillations induced by light delivery to two different, more widely regions of the slice. Notably, the width of the blue light region for this experiment was 600 μm whereas for the other experiment (two light patches to induce oscillations at two locations) it was 300 um. This raises the question of whether synchronization in this experiment vs. during stimulation of two different more widely separated regions are really the same phenomenon. In particular, when the patch of blue light is larger, the distance between the edge of the induced oscillation and the distant recorded cell will be much smaller than the distance between the center of the stimulated region and the "distant" recorded cell.The title and Abstract suggest that the main motivation for this study is identifying the neural circuit that mediates long-range synchronization of oscillations. This is a very interesting topic that is certainly worthy of publication in eLife and the methods used are appropriate. Based on the issues raised above, I'm not sure the authors actually reveal this mechanism. That being said, I am enthusiastic about the direction of this work and think this concern could be addressed by repeating these experiments as a function of distance, and possibly making recordings from pyramidal cells and/or SOM interneurons in the intervening region.

We have amended the schematics in Figure 4 as per the reviewer’s recommendations.

In the experiment in Figure 4, we aimed to confirm the conclusion that SOM cells mediate γ entrainment across the horizontal axis. We originally tried separating the two regions (blue patch and spiking Pyramidal cell/red patch) by distances as long as in Figure 1 (up to ~800um), as the reviewers suggests, however, we found that the net synaptic input to the target pyramidal cell was weak and γ entrainment was not robust compared to that seen in Figure 1 or the shorter edge-to-edge distance that we ultimately used in Figure 4. The best explanation for this is that the strong coherence seen at long distances in Figure 1 is because *both* sites get blue light photo-stimulation, so that interneurons in both patches receive both local input from nearby pyramidal cells (which is relatively stronger and brings them to action potential threshold), and a smaller amount of long-range input from the other blue light patch. Thus we concluded that while the long-range input, on its own, is insufficient to bring interneurons to AP threshold, it is nevertheless sufficient to phase lock the two independently oscillating ensembles into coherence when they are both driven with blue light. Conversely, in the experiment in Figure 4, since only one patch gets direct photo-stimulation with blue light, the interneurons in the red light patch are only receiving distal input, which, when restricted to long-range distances, is insufficient on its own to bring local [SOM] interneurons to action potential threshold. Hence, we used a wider blue light patch that includes both long-range (up to ~950μm) and more proximal regions of L2/3 (~200μm) (see updated Figure 4 and Figure 4 supplement). Taking these constraints into consideration, this configuration tests the necessity of SOM cell activity on the sum of more proximal *and* long-range horizontal input, rather than just long-range input alone. Given the long-range coherence and recruitment we observe in Figure 1, as well as literature on connectivity as a function of distance (Fino and Yuste, 2011; Kato et al. 2017), we surmise that both nearby and long-range inputs contribute to phase coherence in the recorded spiking Pyramidal cell in Figure 4.

Upon suppressing local SOM cells (within ~200 μm), we find coherence is abolished. Critically, this experiment demonstrates that whether it be local SOM cells or distally-projecting SOM cells, it is indeed SOM cells mediating both nearby and long-range γ entrainment. If other interneuron subtypes or direct excitation meaningfully contributed to nearby or long-range entrainment, spike-oscillation coherence upon red light illumination would remain to some extent, yet we see it abolished nearly completely. This result solidifies our core hypothesis that SOM cells mediate long-range γ synchronization, and is discussed more clearly in the revised manuscript.

Finally, the authors should quantify/characterize the frequency of the induced oscillations and be specific about this in the Abstract since "γ" oscillations mean different things to different people.

We have added this information to the Abstract.

[Editors' note: further revisions were requested prior to acceptance, as described below.]

The manuscript has been improved but there are some remaining issues that need to be addressed before acceptance, as outlined below:The reviewers fully support the study and find that the phenomena outlined here are very interesting, important, and novel. However, the mechanistic understanding remains incomplete. Therefore additional experiments are required. The reviewer asks for one set of important experiments: record from SOM cells and pyramidal cells at various distances from a single patch of blue light / ChR2 excitation, and evaluate whether these cells receive long-distance EPSCs, long-distance IPSCs, and/or currents that appear to be gap-junction mediated. If SOM interneurons 400-500 µm away from the blue light patch receive EPSPs but do not show evidence of gap-junction mediated currents (e.g. spikelets) and there are not IPSPs in pyramidal neurons at a similar distance, then this would provide definitive evidence in favor of the author's proposed mechanism.

We are glad the reviewers fully support the study, and we have conducted the proposed experiments. Our data confirms the three predictions mentioned above: First, at 450 µm SOM cells get appreciable (though reduced) excitatory currents. Second, pyramidal cells get even less inhibition. Third, we observed no obvious spikelets or evidence of propagation of subthreshold potentials in SOM cells that would indicate propagations of signals between electrically coupled SOM cells across the retinotopic axis. We discuss these points in detail below.

Reviewer #3:

I completely agree with the authors that:1) The role of SOM interneurons in γ-frequency synchronization has long been underappreciated by the field;2) SOM cells are essential for synchronization in the second experiment (shown in Figure 4).The two key questions, which in my mind remain unanswered, are:1) Does the same mechanism support long-range synchronization in the first experiment (Figure 1) and shorter-range synchronization in the second experiment (Figure 4)?2) In the second experiment, what is the source of SOM-mediated synchronization – is it gap-junction coupling, medium-range SOM projections, or medium range excitatory input to local SOM interneurons?

Fully discussed below, but the new circuit mapping data implies that synchronization in slice beyond 300 µm benefits from convergent excitatory input from distant excitatory ensembles on both sides to drive individual SOM cells. While the extent of horizontal excitatory projections is surely critical, the spatial spread of synchronization is likely expanded by the horizontal spread of SOM axons themselves, as suggested by the reviewer.

Specifically, with regard to #2: the authors argue that gap-junction coupling is not sufficient to extend the range of synaptically evoked inhibition. That may be true, but it doesn't mean that SOM interneurons don't receive subthreshold rhythmic gap junction mediated currents that elicit synchronization.

We agree with this point and have amended the text. We recorded SOM cells in current clamp while stimulating increasingly more distant ensembles of excitatory neurons, but never observed obvious spikelets. This doesn’t rule out a contribution of gap junctions, since spikelets might appear temporally filtered to an extent they are indistinguishable or buried in the large barrages of glutamatergic EPSPs (Hu and Agmon, 2015), and the text has been updated to reflect this. With respect to subthreshold rhythmic gap junction currents, this is also not something we can entirely rule out. However, if subthreshold electrical propagation contributed to SOM cell spike-timing, we might expect that we would observe a filtered version of what a SOM neuron receives for more proximal stimulation (that is, a low pass filtered version of EPSPs). Instead, membrane current traces in SOM neurons show clear periods of baseline/quiescence between what appear to be unitary-like synaptic conductances or EPSPs in response to distal excitation (Figure 4—figure supplement 2); traces during more proximal stimulation show no such quiescent periods. This is not consistent with substantial propagation of subthreshold electrical inputs from proximal to distal SOM cells.

Given the significant off-target effects of all gap-junction blocking drugs (Juszczak and Swiergiel, 2009), future work in mouse lines with connexin knock outs could more thoroughly address this point. Although there is little work on this topic, genetic deletion of connexin36 (Cx36) largely abolishes electrical coupling between cortical interneurons (Deans et al., Neuron, 2001), and thus could be used as a reasonably definitive test for the requirement of electrical coupling in the light evoked γ. These authors have previously presented data (Connors et al., SFN Abstract, 2014, “Roles of electrical synapses in γ-band activity of sensory and association cortex”) that showed that light-ramp evoked γ oscillations in the sensory cortex are observed in Cx36 KO mice, although these γ oscillations were in the barrel cortex and these authors used an AAV with a CamKII promoter, which is not selective to L2/3. However, it should be possible to test the contribution of gap junctions to this γ coherence definitively by electroporating ChR2 into V1 of Cx36 KOs – an interesting experiment we think is appropriate for a future study.

The authors argue against the possibility of medium-range SOM projections, by arguing that eNpHR would not be effective for silencing terminals, based on published data from Mahn et al. I would point out that the situation here is quite different from Mahn et al. – the distance between the edge of the red and blue light patches is only 150 um. Given that the red light seems to inhibit cells ~100 μm away (Figure 4), and that the oscillation extends some distance beyond the edge of the blue light, it seems likely that the red light will directly suppress the activity of some SOM cells involved in the oscillation (not just inhibit their terminals).

We agree with this point as well, and have amended the text. As mentioned above, our data indicate that the spatial spread of L2/3 excitatory axons is critical, but the spatial spread of SOM cell axons themselves are likely to extend the range of synchronization. In Figure 4, as the reviewer points out, some of the drop in synchronization is most likely a result of direct somato-dendritic suppression of SOM cells that lie between the blue and red light zones. The text has been amended to reflect this fact.

So while the phenomena outlined here are very interesting, important, and novel, the mechanistic understanding remains incomplete. I am sympathetic to the experimental constraints noted by the authors. It may not be possible to repeat the second experiment using a smaller blue light patch/larger gap. However, given that this is a relatively short paper that builds on the author's previous work, my opinion is that there should be some test of how synchrony is transmitted by SOM cells. This could be quite minimal – I think simply recording from SOM cells and pyramidal cells at various distances from a single patch of blue light / ChR2 excitation, and evaluating whether these cells receive long-distance EPSCs, long-distance IPSCs, and/or currents that appear to be gap-junction mediated, would be very informative. I.e., if SOM interneurons 400-500 μm away from the blue light patch receive EPSPs but do not show evidence of gap-junction mediated currents (e.g., spikelets), and there are not IPSPs in pyramidal neurons at a similar distance, then this would provide definitive evidence in favor of the author's proposed mechanism.

We have conducted the proposed experiments. The data is presented as Figure 4—figure supplement 2. These experiments show that SOM cells do receive appreciable current as far away as 450 µm. While the absolute amount of excitatory synaptic charge at this distance dropped to ~6% of the maximum, the max synaptic current still averaged 120 ± 90 pA, and several of the recorded SOM cells exhibited depolarizations of 10-20 mV, and occasionally, though rarely, could even be driven to spike. The large depolarizations (and some evoked spiking), despite the reduced currents, are likely due to the relatively high membrane resistance of SOM cells (also note that even 6% of the very strong input seen at 0 µm may still be sufficient input to facilitate synchronization with other active ensembles even if it does not spike the SOM neurons on its own). At these same distances we observed no obvious spikelets (see expanded voltage- and current-clamp traces in Figure 4—figure supplement 2), although these cannot be entirely ruled out. We also quantified inhibitory synaptic input in pyramidal cells as a function of distance. The overall shape of this curve compared to that of synaptic excitation to SOM neurons is similar (consistent with Adesnik and Scanziani, 2010, in the barrel cortex), although it may fall off slightly faster in space as the reviewer predicts. Perhaps more importantly, the absolute amplitude of the inhibitory currents were very small (max 20 ± 10 pA, and sometimes non-existent at 450 µm).

We conclude from these data that the loss of synchronization observed in Figure 4 (entrainment in SOM cell at ~300-400 µm from blue light patch) is likely due to somatodendritic suppression of SOM cells that are mostly located near, but not necessarily in, the red light zone. SOM cells in the red light zone itself are substantially depolarized by horizontal input, and a few of them are even driven to spike, but probably don’t contribute as much as those SOM cells that lie between the two light patches. The Discussion has been updated to reflect this point.

In Figure 1, when we observed γ coherence between ensembles separated by as much as 800 µm, we propose the scenario is somewhat different (and potentially more similar to what is going in the brain during sensory stimulation with multiple stimuli). Here, two spatially separated ensembles of excitatory neurons – each driven independently to oscillate with their own patch of blue light – converge on SOM cells across the horizontal axis, but particularly in between them. The combined input would drive these SOM cells to spike. Thus, SOM cells across the horizontal axis are recruited by summating input from the two sides, and the combined action of all these SOM cells help synchronize the distant ensembles. Despite the absence of observed spikelets or electrically coupled subthreshold potentials, we don’t rule out gap junctional coupling as a contributor to this process, but assessing the specific role of gap junctions will require future work in CXx36 animals.

The long-range transmission of synchrony (potentially) by SOM neurons is the key novel finding that differentiates this from the author's previous work, which is why I think this kind of additional evidence is important. If the authors were to find, for example, contrary to the preceding, that SOM cells at the border of the illuminated region (e.g., 50 μm beyond the edge of the blue light) provide inhibition to pyramidal cells ~200-300 μm away (very plausible given Figure 4 of Fino and Yuste), and SOM interneurons >300 μm away from the illuminated region do not receive significant EPSPs, then this would call into question the authors' interpretation.

As described above, based on the new data, we have appropriately revised our proposed mechanism. All these data are now in the manuscript, and a full discussion of its interpretation has been added to the Discussion. In brief, SOM interneurons >300 μm away from the illuminated region do get significant synaptic input that could facilitate γ entrainment, but we have updated the text to reflect the fact that the spatial spread of SOM neurons’ axons themselves most likely contributes to the propagation of synchrony as well. With these modifications, we feel that the current interpretation as it stands in the revised manuscript is fully supported by the data.